# A sensitive MOKE and optical Hall effect technique at visible wavelengths: insights into the Gilbert damping

Nadav Am-Shalom[1], Amit Rothschild[1], Nirel Bernstein[1], Michael Malka[1], Benjamin Assouline [1], Daniel Kaplan [2], Tobias Holder[2], Binghai Yan[2,3], Igor Rozhansky [4] & Amir Capua [1]✉

It is well known that the anomalous Hall effect displayed in ferromagnets is much stronger than the ordinary Hall effect. Therefore, the optical Hall effect is significantly weaker than the magneto-optical Kerr effect (MOKE) such that it is barely detectable at visible wavelengths. We present a sensitive MOKE technique which is based on large-amplitude modulation of the externally applied magnetic field that is suitable for non-magnetic metals. Using a 440 nm laser, we measure Cu, Au, Al, Ta, and Pt and find partial agreement with the Lorentz-Drude theory implying contributions of the plasma dynamics and interband transitions beyond the approximations of the model. Interestingly, we find that the noise scales with the spin-orbit coupling of the metals. This is manifested by a remarkable correlation between the noise amplitude and the Gilbert damping enhancement associated with these metals. These results suggest that the electromagnetic noise arises from optical interactions with spins that is mediated by the spin-orbit interaction and highlight a possible avenue for measuring the spin-orbit coupling using optical techniques.

In the ordinary Hall effect, the Lorentz force deflects the electrons and the transverse Hall voltage arises. When the measurement is carried out in magnetic materials, the anomalous Hall effect (AHE) occurs with an anomalously large Hall voltage that saturates with the applied magnetic field[1–4]. The magneto-optic Kerr effect (MOKE) can be regarded as the optical analogue of the AHE. It is characterized by an anomalously large transverse polarization that saturates with the externally applied magnetic field such that it is sensitive even to a single atomic layer of a magnetic element[5–8]. While the MOKE is applied routinely, measurement of its non-magnetic counterpart at visible light wavelengths is more challenging. This is known as the optical Hall effect (OHE)[9,10] as illustrated schematically in Fig. 1a.

The OHE has been primarily measured at THz and infrared frequencies where the effective electronic displacement is larger. Such studies were carried out in semiconductors[11,12] and 2D materials[13–15]

while exceptionally large rotation angles were found in topological insulators[16,17]. At visible light wavelengths, a variety of techniques were applied while much of the research concentrated on Cu, Au, and Ag due to their similar band structure[18]. In the early studies by Stern and McGroddy[19], multiple reflections extended the optical interaction length and enhanced the sensitivity. Later, the sensitivity was achieved by modulating the optical polarization state of the incident beam which enabled measurements of the effect in the near UV spectrum and of the plasma resonance response[20,21]. Studies in thin films of Au and Cu at very high magnetic fields found the Hall frequency to be consistent with band theory predictions[22]. These observations were obtained by simultaneously measuring the Faraday rotation and circular dichroism[3]. Further connection between the OHE and the band structure physics was presented in ref. 10, where the complete optical conductivity tensor of the three metals was extracted over the entire

[1]Insititute of Electrical Engineering & Applied Physics, The Hebrew University of Jerusalem, Jerusalem, Israel. [2]Department of Condensed Matter Physics, Weizmann Institute of Science, Rehovot, Israel. [3]Department of Physics, Pennsylvania State University, University Park, PA, USA. [4]National Graphene Institute, University of Manchester, Manchester, UK. ✉e-mail: amir.capua@mail.huji.ac.il

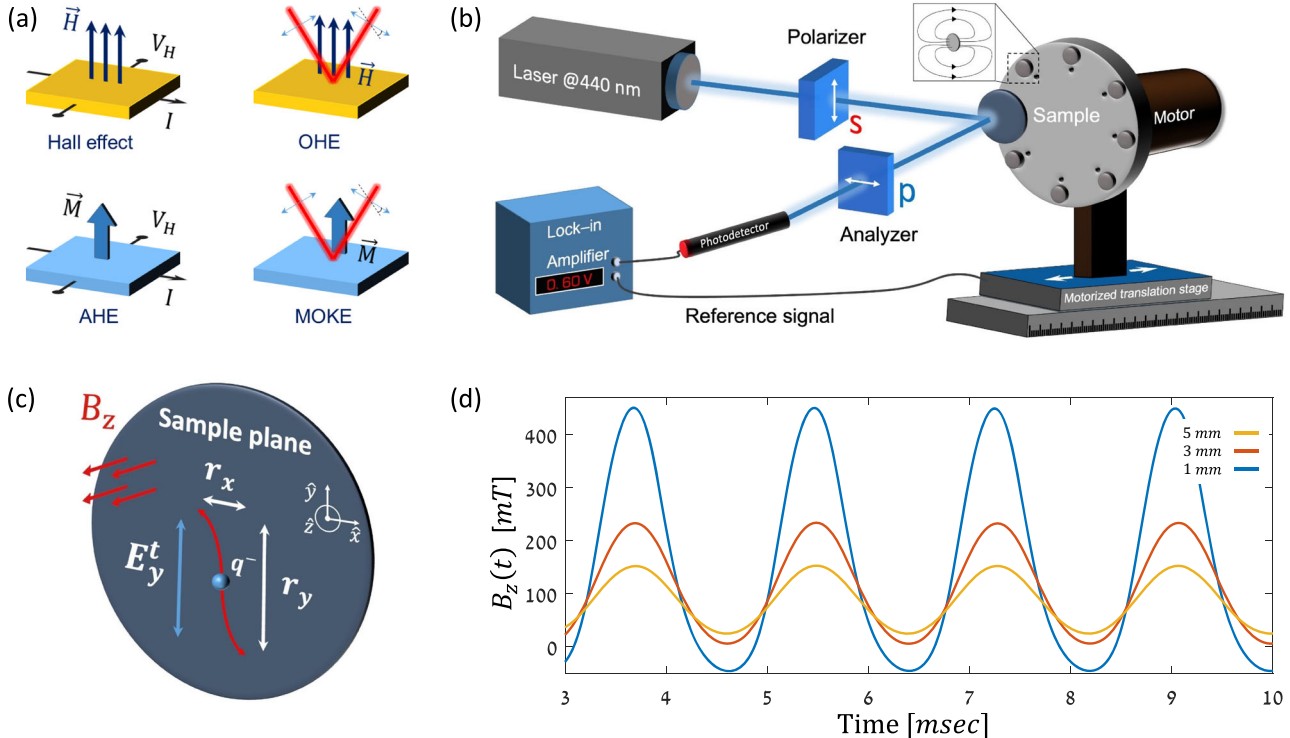

**Fig. 1 | Schematic illustration of the OHE and the experimental setup. a** Analogy between the ordinary Hall effect, AHE, MOKE, and OHE. **b** Ferris MOKE experimental setup. Inset: illustration of the magnetic field lines of a single magnet. **c** Schematic illustration of the induced transverse polarization in $\hat{x}$ due to the linearly polarized optical field in $\hat{y}$ and the externally applied magnetic field, $\boldsymbol{B_z}$. **d** Measured temporal profiles of $\boldsymbol{B_z(t, l)}$ for $\boldsymbol{l = 1, 3}$, and $\boldsymbol{5\ mm}$.

spectral range of $0.8 - 5.5\ eV$. These measurements were complemented by a novel ab-initio band structure theory derived within the limit of the spin-polarized relativistic linear-muffin-tin-orbital approximation and revealed that the differences between the measured magneto-optical responses of the three samples originated in the different positions of the energy bands.

The sensitivity to the Kerr signal can be increased in a straightforward manner by modulating the external magnetic field. However, when electromagnets are used, this can be achieved only at non-practical low rates and amplitudes. Here, we present a MOKE technique in which a large-amplitude modulation at a sufficiently high frequency is achieved using permanent magnets that are placed on a rotating disc. Consequently, the sensitivity is significantly increased as compared to a more conventional light-modulated MOKE such that the OHE becomes measurable at optical frequencies. The approach resembles the Ferris ferromagnetic resonance (FMR) technique[23,24] operating with RF excitations in which the large amplitude magnetic field modulation leads to a proportionally large FMR signal representing the RF absorption in the sample. Hence, we refer to it as the *Ferris MOKE*.

The technique is demonstrated on thin films of Cu, Au, Al, Ta, and Pt and the responses are compared to the Lorentz-Drude model[3,9,10,18] that also accounts for the evanescent field in the metal within the planar charge oscillations approximation. We find partial agreement indicating possible contributions of the plasma dynamics and interband transitions beyond the Lorentz-Drude approximation. Interestingly, the responses readily illustrate that the Ferris MOKE measurements are significantly noisier in the heavier metals as compared to the lighter ones. This implies a connection with their strong spin-orbit coupling (SOC). In ferromagnetic materials, SOC stands behind the loss of spin angular momentum to the lattice that is quantified by the Gilbert damping parameter, $\alpha$. Moreover, heavy metals are known to be excellent absorbers of spin angular momentum

and enhance $\alpha$ when placed in proximity to a ferromagnet (FM). Therefore, we compare the standard deviation of the noise signal and the $\alpha$ enhancement, $\alpha_{sp}$, induced by these metals by spin pumping[25] and find very good correlation. These observations suggest that the noise arises from optical interactions with spins mediated by SOC.

## Results

The experimental setup is based on the polar MOKE geometry illustrated in Fig. 1b. It consists of two orthogonally aligned linear polarizers having an extinction ratio, $ER$, better than $1:10^6$ and a $40\ mW$ CW laser operating at $440\ nm$. The modulated external magnetic field $B_z(t)$ is generated using permanent magnets placed on a spinning disc having an out-of-plane magnetic field along $\hat{z}$ required for the polar geometry. A motorized translation stage controls the sample-magnet distance $l$ while the voltage-drop on the photodetector $V_{PD}$ is measured using a lock-in amplifier. The incident electrical field is linearly polarized in $\hat{y}$ ($s$-polarization) and induces an electronic displacement along $\hat{x}$ which generates the $p$-polarization by the Lorentz force as illustrated schematically in Fig. 1c. An example of the generated $B_z(t, l)$ for $l = 1, 3$, and $5\ mm$ is presented in Fig. 1d. At a given position, $B_z(t, l)$ can be described by the sum of an all-positive sine wave of amplitude $B_{AC}(l)$, a small DC bias level $b_0(l)$, and additional higher harmonics such that $\qquad B_z(l, t) = b_0(l) + \frac{1}{2}B_{AC}(l) \cdot (1 + \sin(\omega_{mod}t)) +$ higher harmonics with $\omega_{mod}$ being the modulation angular frequency. The amplitude of the higher harmonics is small and their contribution to the measured signal is further suppressed by the lock-in amplifier. More details of the experimental setup are presented in 'Methods' section.

We start by comparing the Ferris MOKE measurement of Py to a more conventional implementation having a static external magnetic field, $B_{DC}$, and an on-off modulated laser beam (more details in 'Methods'). $B_{DC}$ was applied by fixing the rotating disc at the maximal field of one of its magnets and was varied by scanning over $l$. Both experiments were carried out with the same optical alignment and

(a)

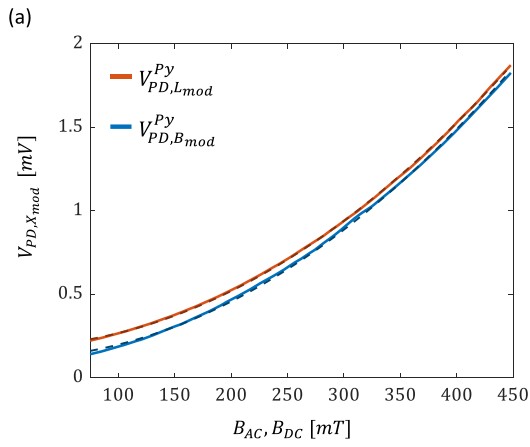

(b)

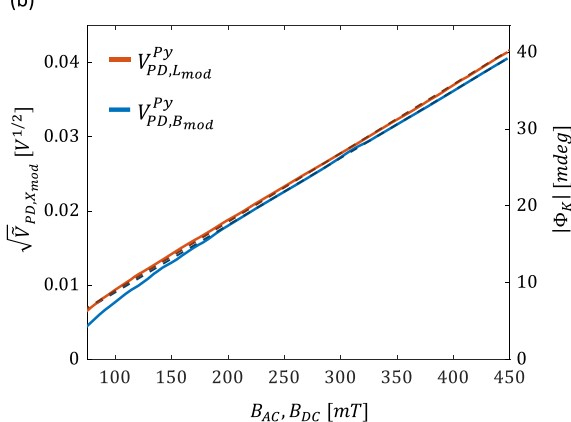

**Fig. 2 | Comparison between the Ferris MOKE and light-modulated MOKE measurements in Py. a** $V_{PD,B_{mod}}$ (solid blue) and $V_{PD,L_{mod}}$ (solid red) vs. $\boldsymbol{B_{AC}}$ and $\boldsymbol{B_{DC}}$, respectively. Dashed black lines represent the parabolic fit. **b** $\sqrt{\tilde{V}^{Py}_{PD,B_{mod}}}$ and $\sqrt{\tilde{V}^{Py}_{PD,L_{mod}}}$ vs. $\boldsymbol{B_{AC}}$ and $\boldsymbol{B_{DC}}$, respectively, and the corresponding magnitude of the complex Kerr angle $|\boldsymbol{\Phi_K}|$. Dashed lines represent linear fit for data in the range of $\boldsymbol{B_{AC}, B_{DC} > 200mT}$.

lock-in amplifier settings. The in-plane magnetized film was $50\,nm$ thick and was capped with $2.5\,nm$ of TaN (more details in 'Methods' section). The measured photodiode voltages $V^{Py}_{PD,B_{mod}}$ and $V^{Py}_{PD,L_{mod}}$ of the Ferris and the conventional MOKE experiments, respectively, are presented in Fig. 2a (sub-indices $B_{mod}$ and $L_{mod}$ indicate quantities measured with $B_z$- and laser- modulations, respectively, and upper indices indicate the probed material). A quadratic dependence on $B_{AC}$ and $B_{DC}$ is measured in both cases and is typical of the maximal extinction cross-polarized arrangement[26]. At low $B_{AC}$ and $B_{DC}$, $V^{Py}_{PD,B_{mod}}$ approaches lower values as compared to $V^{Py}_{PD,L_{mod}}$. In contrast, at high magnetic fields both signals display a similar dependence on $B_{AC}$ and $B_{DC}$ besides a constant shift of $V^{Py}_{PD,L_{mod}}$ to higher values. These observations indicate a higher noise floor of the light-modulated MOKE that stems from stray laser light and imperfections of the polarizers. Since the transverse polarization is proportional to $B_z$ and the detected photocurrent is proportional to the optical power, $V_{PD}$ can be fitted to $V_{PD,X_{mod}} = a \cdot x^2 + b$ with $x = B_{AC}, B_{DC}$. Accordingly, we define $\tilde{V}_{PD,X_{mod}} \triangleq a \cdot x^2$ where $\sqrt{\tilde{V}_{PD,X_{mod}}}$ is proportional to the amplitude of the generated transverse electrical field. To mitigate the contributions of the noise floor, the fitting procedure was carried out in the range of $B_{AC}, B_{DC} > 200\,mT$. Figure 2b presents $\sqrt{\tilde{V}_{PD,X_{mod}}}$ illustrating a linear dependence on $B_{AC}$ ($B_{DC}$) with $a^{1/2,Py}_{B_{mod}} = 90.5\,mV^{1/2}/T$ ($a^{1/2,Py}_{L_{mod}} = 91.2\,mV^{1/2}/T$).

$V_{PD}$ measures the transverse $p$-polarization, which originates from the Kerr rotation, $\theta_k$, and ellipticity, $\epsilon_k$. These parameters can be measured separately by using a balanced detection scheme that includes a controllable retarder as described by Ortiz et al. and Gomez et al. in refs. [27–29]. In the maximal extinction cross-polarized detection line, only the magnitude of the complex magneto-optical Kerr angle, $\Phi_K = \theta_k + i\epsilon_k$ can be measured. $|\Phi_K|$ indicates the strength of the MOKE and is useful for quantitative comparison with other reports and materials. To describe the relation between $|\Phi_K|$ and $V_{PD}$, we use the Jones calculus and follow the derivation by Gomez et al.[27] (see Supplementary Note 1). The Jones matrix of the sample is given by: $\boldsymbol{S} = \begin{bmatrix} r_{pp} & r_{ps} \\ r_{sp} & r_{ss} \end{bmatrix}$. $r_{pp}$ and $r_{ss}$ are the complex Fresnel reflection coefficients for the $s$ and $p$ polarizations, respectively, and the MOKE response is manifested by the off-diagonal elements, $r_{ps}$ and $r_{sp}$. For an $s$-polarized incident field, $\Phi_K = \frac{r_{ps}}{r_{ss}}$. In the maximal extinction cross-polarized configuration, $V_{PD} = \frac{1}{2} \cdot c\epsilon_0\rho_0|E^i_0|^2 \cdot |r_{ps}|^2$, where $E^i_0$ is the amplitude of the incident laser field, $\rho_0$ is the optical responsivity, and

$c$ and $\epsilon_0$ are the speed of light and permittivity in vacuum, respectively. $V_{PD} \propto |r_{ps}|^2$, therefore, it can be detected at the fundamental tone, $\omega_{mod}$, or at $2\omega_{mod}$ (see Supplementary Note 2). Since $|r_{ps}| \ll |r_{ss}|$, $V^0_{PD}$ corresponding to the measurement of $|r_{ss}|$, was determined by placing the photodiode immediately after the sample (see 'Methods'). Following a calibration process (see Supplementary Note 1), $|\Phi_K|$ was extracted and is also presented in Fig. 2b. At the maximal field, $|\Phi_K| \sim 20\,mdeg$ and is comparable to previously reported values[30].

Often, $V_{PD}$ is found to be linear in the applied magnetic field, $B$, whereas, here, it is quadratic in $B$ (for simplicity we use the general notation $B$ rather than $B_{DC}, B_{AC}$). The relationship between $V_{PD}$ and $B$ depends on the implementation of the detection line, the precise angle of the analyzer, and the type of modulation. For an analyzer that deviates from perfect extinction by an angle $\phi$, $V_{PD}$ is given by:

$$V_{PD} = \frac{1}{2} \cdot c\epsilon_0\rho_0\left|E^i_0\right|^2 \cdot \left[\sin^2\phi\left|r_{ss}\right|^2 + \cos^2\phi\left|r_{ps}\right|^2 \right.$$
$$\left. + 2\cos\phi\sin\phi \cdot \mathrm{Re}\left\{r_{ss}r^*_{ps}\right\}\right] \tag{1}$$

(see Supplementary Note 1). $r_{ss}$ is independent of $B$ whereas $r_{ps}$ is linear in $B$ since the Kerr rotation is linear in the induced magnetization. Therefore, in the $B$-modulation case, only the second and third terms will be detected. Of the two terms, for a slight deviation of $\phi$ from $0°$, the third intermixing term will dominate since $|r_{ss}| \gg |r_{ps}|$. Consequently, the response will become linear in $B_{AC}$. In the case of laser-modulation, all terms will be detected by the lock-in amplifier. Therefore, $V_{PD}$ will additionally be shifted by the leading $B$-independent term which increases rapidly with $\phi$. Measurements of the $B$- and $L$- modulation for slight deviations of $\phi$ are presented in Supplementary Note 3. These measurements readily illustrate that the transition from a parabolic response to linear relationship is reached already for $\phi \approx 60\,mdeg$. When a balanced detector is used to detect the Kerr rotation[27,28], only the third intermixing term is measured, always resulting in a linear relationship on $B$ independent of the modulation type (see Supplementary Note 4). The agreement between Eq. (1) and the additional measurements presented in Figs. S3 and S4 indicates that the quadratic relationship stems from the MOKE-induced polarization rotation. It is possible that additional effects take place such as magnetostriction which couples the mechanical and spin degrees of freedom. This was demonstrated by Jiang et al.[31] in a

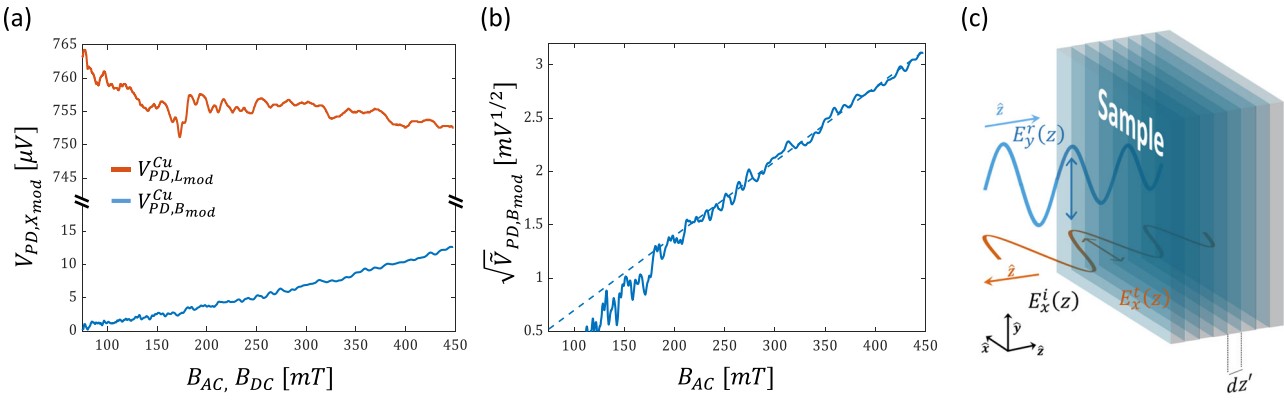

**Fig. 3 | Optical Hall measurements of Cu. a** Measurement of $V_{PD,B_{mod}}^{Cu}$ (blue solid line) and $V_{PD,L_{mod}}^{Cu}$ (red solid line) vs. $B_{AC}$ and $B_{DC}$, respectively. **b** Measurement of $\sqrt{\widetilde{V}_{PD,B_{mod}}^{Cu}}$ vs. $B_{AC}$. Dashed blue line represents a linear fit for data in the range of $B_{AC} > 200\,mT$. **c** Schematic illustration of the modeled interaction. The transmitted field, $E_y^t(z)$, propagates in the metal and generates the reflected field $E_x^r$.

suspended 2D magnetic CrI₃ membrane by correlating the magnetic circular dichroism signal with the mechanical resonance frequency. In contrast, the sample in our case is much thicker and was deposited directly on a rigid substrate, therefore, the elasticity is significantly smaller.

The lower noise floor of the Ferris MOKE calls to examine the OHE response in normal metals (NMs). To this end, we first compare the Ferris MOKE and light-modulated MOKE responses of a 50 $nm$ Cu film. The film was capped as well with 2.5 $nm$ of TaN to prevent native surface oxidation and its thickness is much larger than the optical penetration depth. Figure 3a presents the measured $V_{PD,B_{mod}}^{Cu}$ and $V_{PD,L_{mod}}^{Cu}$. The high noise floor of the light-modulated MOKE prevents detecting any meaningful response. In contrast, $V_{PD,B_{mod}}^{Cu}$ is much lower and reveals a clear parabolic dependence on $B_{AC}$. Figure 3b presents $\sqrt{\widetilde{V}_{PD,B_{mod}}^{Cu}}$ which is linear in $B_{AC}$ at the higher fields yielding $a_{B_{mod}}^{1/2,Cu} = 7.88\,mV^{1/2}/T$. The comparison of Cu and Py is of much interest since Py consists of 81% Ni which is the nearest neighbor to Cu in the periodic table. Hence, Ni and Cu share a variety of properties including the crystal structure, a close atomic weight, and a weak SOC. Here, we find the ratio $a_{B_{mod}}^{1/2,Py}/a_{B_{mod}}^{1/2,Cu} = 11.48$ which agrees with the general notion that the Hall voltage in DC transport measurements is ~10 times larger in FM as compared to NMs[4].

To the lowest order, the OHE can be modeled by considering the un-bound charge carriers within the framework of Newtonian physics[9,32]. To this end, we determine the transverse reflected electrical field $E_x^r$ generated by the electrons' displacement $\mathbf{r}$. In metals, the cyclotron frequency is typically in the microwaves and is much smaller than the optical frequency $\omega_{opt}$ such that $r_x \ll r_y$. For a plasma excitation having a resonance frequency of $\omega_p$ and a Drude scattering time, $\tau$, the equation of motion is $m^* \ddot{\mathbf{r}} = q(\mathbf{E}^t - \dot{\mathbf{r}} \times \hat{\mathbf{z}} \cdot B_z) - m^* \omega_p^2 \mathbf{r} - \frac{m^*}{\tau} \dot{\mathbf{r}}$, where $\mathbf{E}^t = E_y^t(z)e^{-i\omega_{opt}t}\hat{\mathbf{y}}$ is the driving evanescent electrical field in the metal, and $m^*$, and $q$ are the effective mass of the electron, and the elementary charge, respectively. Core-level transitions are neglected as they occur in the X-ray range. In steady state, $\mathbf{r} = (r_x\hat{\mathbf{x}} + r_y\hat{\mathbf{y}})e^{-i\omega_{opt}t}$ with

$$r_x = \frac{i\omega_{opt}q^2}{m^{*2}\left(\omega_p^2 - \omega_{opt}^2 - i\omega_{opt}/\tau\right)^2 - \omega_{opt}^2 q^2 B_z^2} \cdot E_y^t(z) \cdot B_z. \quad (2)$$

As $\mathbf{E}^t$ penetrates the film, it excites the transverse polarization which adds to $E_x^r$ as illustrated schematically in Fig. 3c. Along $\hat{\mathbf{z}}$, $E_y^t$ is expressed by $E_y^t(z) = t_{12} \cdot E_0^i e^{ink_0 z}$ where $n$ is the complex index of

refraction, $t_{12}$ is the Fresnel transmission coefficient, and $k_0 = \omega_{opt}/c$. Following the planar charge oscillations approximation[33] and Eq. (2), the contribution of an infinitesimal sheet of polarization located at $z'$ to the reflected field at the interface $E_x^r(z=0)$ can be expressed by $dE_x^p(z=0,t,z') = t_{21}\frac{n_i \cdot dz' \cdot q}{2\epsilon_0 c/n} \cdot \frac{\partial r_x}{\partial t}$, where $t_{21}$ is the metal-to-air Fresnel transmission coefficient. Accordingly, $E_x^r(0) = \int_{z'=0}^{\infty} dE_x^p(0,t,z')$ resulting in:

$$V_{PD} = C_0 \cdot \left(\frac{\omega_{opt}q^3 E_0^i}{2\epsilon_0} \cdot \frac{n_i}{m^{*2}\left(\omega_p^2 - \omega_{opt}^2\right)^2 - \omega_{opt}^2 q^2 B_z^2}\right)^2 B_z^2 \cdot |A(n)|^2 \quad (3)$$

where $A(n) \triangleq \frac{n}{n'} \cdot \frac{n}{(1+n)^2}$ and accounts for all index-related contributions such as reflectance and absorption of the material. $C_0$ is a constant factor identical for all samples that depends on a variety of parameters including the quantum efficiency of the photodiode, the impedance of free space, attenuation of the polarizers, responsivity of the photodiode, and the electrical characteristics of the detector. In Eq. (3) we assume that the electrons are decelerated before electron scattering and decoherence takes place since $\tau$ is of few hundreds of femtoseconds. Using Eq. (3), we arrive to the expression $|\Phi_K| = \left|\frac{n^2}{n'(1-n^2)}\right|\left(\frac{\omega_{opt}q^3}{2\epsilon_0} \cdot \frac{n_i}{m^{*2}\left(\omega_p^2 - \omega_{opt}^2\right)^2 - \omega_{opt}^2 q^2 B_z^2}\right)B_z$ which shows the expected linear relationship between the Kerr rotation and $B_z$ for optical frequencies below the plasma frequency.

We evaluate the Lorentz-Drude theory by comparing the model to the OHE responses of Cu, Au, Al, Ta, and Pt. To this end, additional 50 $nm$ films were grown. X-ray diffraction data indicated that the Pt, Al, Au, and Cu samples grew in the preferred (111) crystal orientation while the Ta film was grown in the $\beta$-phase. The surface morphology was characterized using atomic force microscopy. The RMS surface roughness values, $R_q$, of the samples were $R_q^{Pt} = 3.84$ Å, $R_q^{Ta} = 2.7$ Å, $R_q^{Al} = 35$ Å, $R_q^{Au} = 5.5$ Å, and $R_q^{Cu} = 7.1$ Å indicating proper film quality. The measured resistivities were characterized by DC Hall transport and were $\rho^{Pt} = 19.2$, $\rho^{Ta} = 173$, $\rho^{Al} = 24.1$, $\rho^{Au} = 3.9$, and $\rho^{Cu} = 3.3$ $\mu\Omega \cdot cm$. These values are consistent with previous reports while $\rho^{Ta}$ further indicates that the Ta film was grown in the $\beta$-phase. Film characterization data is available in Supplementary Note 5.

The OHE measurements of the additional films are presented in Fig. 4a and display a quadratic dependence on $B_{AC}$ as well. Generally, the quadratic regime is beneficial as it probes solely the off-diagonal magneto-optical response and has a lower noise figure, unlike the

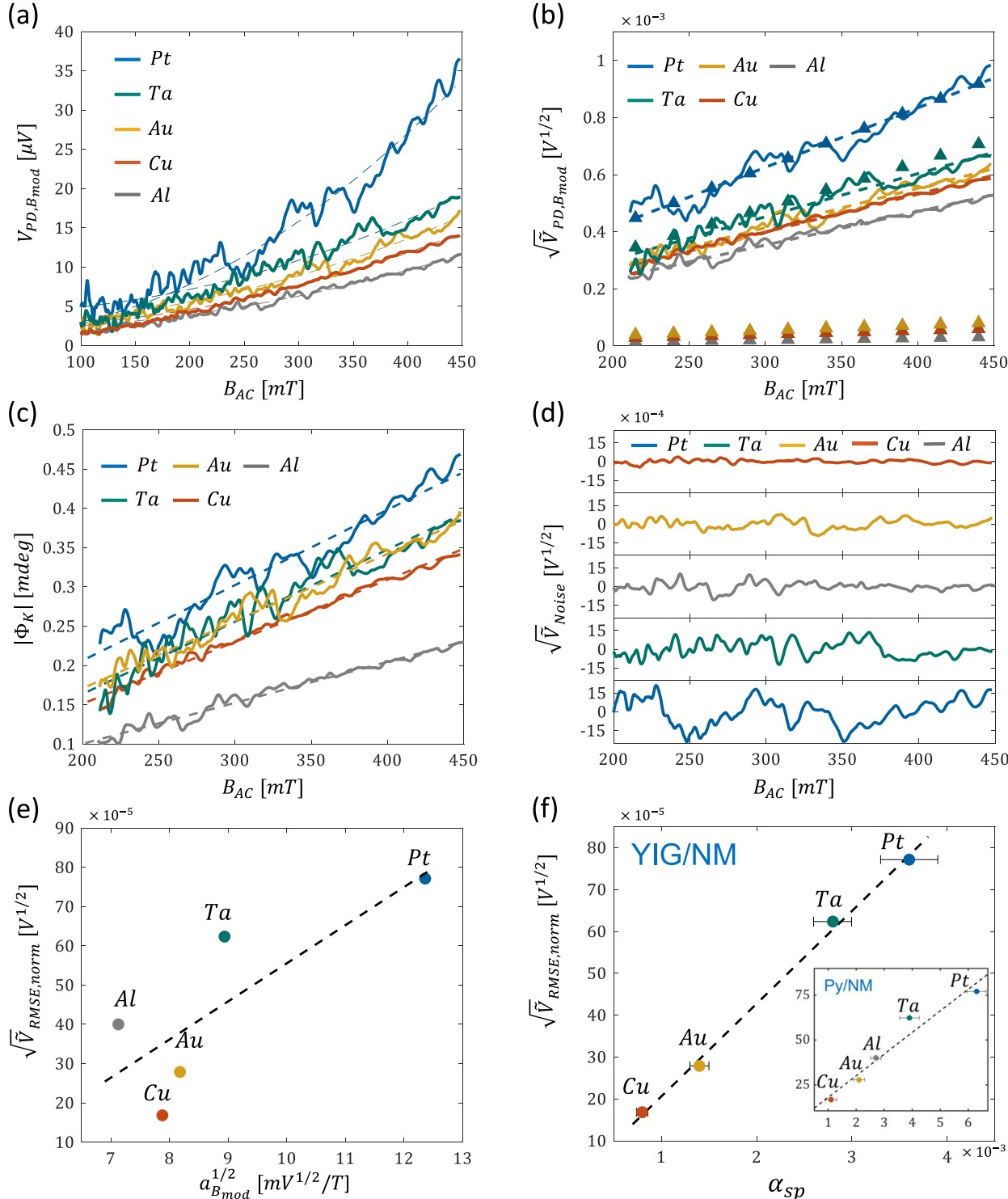

**Fig. 4 | Ferris MOKE measurements of Cu, Au, Al, Ta, and Pt. a** Measured $V_{PD,B_{mod}}$. Dashed lines represent the parabolic fit. **b** Measured $\sqrt{\tilde{V}_{PD,B_{mod}}}$ vs. $B_{AC}$ (solid lines). Dashed lines represent the linear fit of $\sqrt{\tilde{V}_{PD,B_{mod}}}$. Close triangles represent the theoretically calculated $\sqrt{\tilde{V}_{PD}}$. **c** $|\Phi_K|$ of Pt, Ta, Al, Au, and Cu. **d** Extracted $\sqrt{\tilde{V}_{Noise}}$ traces as a function of $B_{AC}$. **e** $\sqrt{\tilde{V}_{RMS,norm}}$ as a function of $a_{B_{mod}}^{1/2}$.

**f** $\sqrt{\tilde{V}_{RMS,norm}}$ as a function of $\alpha_{sp}^{YIG}$. Values were taken from ref. 25. Inset: noise-damping correlation in Py/NM bilayers. In **e**, **f** black dashed line represents a linear fit. In all panels, traces of Cu, Au, Al, Ta, and Pt are indicated by red, yellow, gray, green, and blue colors, respectively.

linear regime, which results from intermixing with the primary polarization $r_{ss}$. The ability to detect the quadratic term depends on the strength of the magneto-optical effect relative to the detection noise. The latter is affected by the amount of light reaching the detector, which depends on the extinction ratio of the polarizers and the

analyzer-polarizer arrangement. Following the derivation by Jin et al. for the optical contrast in a wide-field sensitive magneto-optical imaging system[34], Supplementary Note 6 presents a detailed analysis of the shot noise in typical FMs and NMs and different *ER* and $\phi$ angles. This analysis indicates that maximal extinction ($\phi = 0°$) with high *ER*

**Table 1 | $m^*$, $n_i$, and $n$ values for Cu, Au, Al, Ta, and Pt in addition to the experimentally reported, calculated, and fitted $\omega_p$ values**

| | $m^*/m_0$ | $\omega_p (\times 10^{16})\ [rad/s]$ | | | $n_i\ (\times 10^{22})\ [cm^{-3}]$ | $n$ | $|A(n)|^2$ |
|---|---|---|---|---|---|---|---|
| | | exp. | calc. | fit | | | |
| Cu | 1.3 | 1.33 | 1.74 | 1.01 | 12.4 | $1.24 + i2.34$ | 0.0634 |
| Au | 1.1 | 1.34 | 1.55 | 0.98 | 8.3 | $1.15 + i1.89$ | 0.0725 |
| Al | 0.63 | 2.2 | 3.1 | 1.58 | 19.5 | $0.59 + i5.32$ | 0.0304 |
| Ta | 0.8 | 0.94 | 1.38 | 0.91 | −4.81 | $2.84 + i2.14$ | 0.0337 |
| Pt | 2.05 | 0.82 | 2.03 | 0.82* | 26.1 | $1.84 + i3.18$ | 0.0407 |

$n_i$ values were extracted from transport measurements, while $m^*$ and $\omega_p$ (exp.) values were taken from refs. [38–43]. $m_0$ is the electron's rest mass. *$C_0$ value was chosen to fit the response of Pt.

are necessary to measure the quadratic response in NMs. This is also expected to apply to atomically thin 2D ferromagnets, such as those studied in refs. [7,8,35,36], where the magneto-optical response is significantly weaker as compared to that of 3D ferromagnets.

Figure [4]b presents $\sqrt{\tilde{V}_{PD,B_{mod}}}$ data which are linear in $B_{AC}$. Accordingly, $a_{B_{mod}}^{1/2,Al} = 6.99\,mV^{1/2}/T$, $a_{B_{mod}}^{1/2,Au} = 8.18\ mV^{1/2}/T$, $a_{B_{mod}}^{1/2,Ta} = 8.94\ mV^{1/2}/T$, and $a_{B_{mod}}^{1/2,Pt} = 12.36\ mV^{1/2}/T$ are extracted. Figure [4]c presents the OHE measurements in terms of $|\Phi_K|$ revealing rotation angles in the range of $10^{-1}\ mdeg$ which are comparable to the previously reported values[10,19,20,37] (see Supplementary Note 1 for detailed comparison). To calculate the theoretical $V_{PD}$, $n_i$ values were determined from the electrical DC Hall measurements and are summarized in Table [1] (see 'Methods' section for more details). The negative $n_i$ of Ta stems from its hole-like charge transport which cannot be sensed optically. Values for $\omega_p$, $n$, and $m^*$ were extracted from the experimental data of refs. [38–43]. and are also summarized in Table [1]. Figure [4]b presents the calculated $\sqrt{V_{PD}}$ data indicated by closed triangles where $C_0$ was arbitrarily chosen in accordance with the response of Pt. Since $\omega_{opt}^2 q^2 B_z^2 \ll m^{*2}\left(\omega_p^2 - \omega_{opt}^2\right)^2$ for all samples, the calculated $\sqrt{V_{PD}}$ is essentially linear in $B_z$. It is seen that the Lorentz-Drude model correctly predicts the relative OHE response of Ta but fails to do so simultaneously for Au, Cu, and Al. Nevertheless, it seems that the relative OHE of Au, Cu and Al is predicted correctly. The Pt-Ta and Au-Cu-Al samples are distinguished by $\omega_p$ suggesting that a more detailed description of the connection between the plasma dynamics and the OHE is required to properly account for the measured results. Additionally, the Lorentz-Drude model accounts only for the intraband transitions, whereas interband transitions are also known to have a significant effect. Uba et al. investigated in detail the role of 3d and 5d interband transitions in Cu, Au, and Pt[10,37] which enabled to explain the spectral features of the magneto-optic response. For example, the interband transitions in Cu and Au take place at $\sim 2.1 - 2.4\,eV$ and $\sim 2.5\,eV$, respectively, which lie close to the laser excitation energy of $2.8\,eV$ used in our experiments. Rather than using the previously reported values, $\omega_p$ can be estimated according to the parabolic band approximation by $\omega_p = \sqrt{n_i q^2 / \epsilon_0 m^*}$. Table [1] summarizes the calculated $\omega_p$ based on the measured $n_i$ and the reported $m^*$ values. The calculated $\omega_p$ differ significantly from the experimentally measured $\omega_p$. The deviation stems from the actual band structure that is non-parabolic and anisotropic[44]. Alternatively, for a given $C_0$, $\omega_p$ can be extracted from the measured OHE data. The fitted $\omega_p$ are also presented in Table [1] readily illustrating that the extracted $\omega_p$ values are closer to the previously reported values.

Operation at maximal extinction turns out to be advantageous in terms of the noise introduced to the measurements. The low intensity reaching the detector mitigates the detection shot noise, and the

measurement resolution increases which is well-suited for detecting weak noise signals (see Supplementary Note 7 for further considerations). Figure [4]a reveals an interesting phenomenon related to the noise. It is readily seen that the noise amplitude in the heavier Ta and Pt metals is greater than in the lighter Cu metal. To examine this behavior, the noise signal, $\sqrt{\tilde{V}_{Noise}}$, was extracted by subtracting the linear fit from $\sqrt{\tilde{V}_{PD,B_{mod}}}$ and is presented in Fig. [4]d. It is seen that the noise ascends in the order: Cu → Au → Al → Ta → Pt. This trend is described quantitatively by the root mean square (RMS) of each trace, $\sqrt{\tilde{V}_{RMS}}$, and is summarized in Table [2]. Additionally, to account for the differences in the absolute optical power reaching the photodetector, $\sqrt{\tilde{V}_{RMS}}$ was normalized to $|A(n)|$ and is also indicated in the table by $\sqrt{\tilde{V}_{RMS,norm}}$. Apart from Al, $\sqrt{\tilde{V}_{RMS,norm}}$ seems to correlate with the strength of the OHE response. This is illustrated by plotting $\sqrt{\tilde{V}_{RMS,norm}}$ vs. $a_{B_{mod}}^{1/2}$ in Fig. [4]e. The correlation with the OHE responses implies that the noise is generated by processes such as carrier-carrier and carrier-lattice scattering. However, there are two issues with this picture. Firstly, $\omega_{opt} \gg 1/\tau$, therefore, Drude scattering is understood to be negligible. Secondly, this picture suggests that the noise should increase with $B_{AC}$ which is evidently not the case. Therefore, we conclude that the noise is attributed to a different mechanism. Heavy metals are characterized by large SOC. It seems that the noise amplitude scales with the SOC of these metals as well. SOC is responsible for the transfer of angular momentum from the spin to the lattice. In magnetic materials, it stands behind the loss of spin angular momentum described by $\alpha$. It is well known that $\alpha$ is enhanced when NMs are introduced at the interface of magnetic systems. The enhancement of $\alpha$ by Cu, Au, Ta, and Pt was systematically studied in $Y_3Fe_5O_{12}$ (YIG)/NM bilayers in ref. [25]. These films were grown as well by DC magnetron sputtering on $Gd_3Ga_5O_{12}$ substrates. The YIG and the NM layers were $20\,nm$ and $5\,nm$ thick, respectively, resulting in saturation magnetization of $1.4 \times 10^5\,A/m$. In ref. [25], $\alpha$ was quantified by spin-pumping measurements from the inverse spin Hall effect voltage. From these measurements, the enhancement of $\alpha$ by the NMs, $\alpha_{sp}^{YIG}$, was extracted by subtracting the intrinsic $\alpha$ of YIG from the measured $\alpha$ of the bilayer. This data is also summarized in Table [2]. A significantly better correlation between $\alpha_{sp}^{YIG}$ and $\sqrt{\tilde{V}_{RMS,norm}}$ is readily seen as illustrated in Fig. [4]f. An excellent linear fit is found having an R-square determination parameter of 0.997 further supporting the SOC origin of the noise. In contrast, the R-square value of the fit to $a_{B_{mod}}^{1/2}$ in Fig. [4]e is only 0.64. It is possible that SOC also affects $a_{B_{mod}}^{1/2}$, however, Uba et al. showed that the large SOC of Au, for example, has little effect on its OHE response[10].

YIG exhibits extremely low $\alpha$ which is well-suited for probing the losses of spin angular momentum in the adjacent NM. It is possible that

**Table 2 | Noise correlation data.** $\sqrt{\tilde{V}}_{RMS}$, $\sqrt{\tilde{V}}_{RMS,norm}$, $a^{1/2}_{B_{mod}}$, $\alpha^{YIG}_{sp}$, and $\alpha^{Py}_{sp}$ values for the different samples

|  | $\sqrt{\tilde{V}}_{RMS}$ $\left(\times 10^{-5}\right)\left[V^{1/2}\right]$ | $\sqrt{\tilde{V}}_{RMS,norm}$ $\left(\times 10^{-5}\right)\left[V^{1/2}\right]$ | $a^{1/2}_{B_{mod}}$ $\left(\times 10^{-3}\right)\left[V^{1/2}/T\right]$ | $\alpha^{YIG}_{sp}$ $\left(\times 10^{-3}\right)$ | $\alpha^{Py}_{sp}$ $\left(\times 10^{-3}\right)$ |
|---|---|---|---|---|---|
| Cu | 4.24 | 16.83 | 7.88 | 0.8 ± 0.06 | 1.1 |
| Au | 7.51 | 27.89 | 8.18 | 1.4 ± 0.1 | 2.1 |
| Al | 8.89 | 42.27 | 7.13 | - | 2.7 |
| Ta | 11.45 | 62.35 | 8.94 | 2.8 ± 0.2 | 3.9 |
| Pt | 15.56 | 77.14 | 12.36 | 3.6 ± 0.3 | 6.3 |

$\alpha^{YIG}_{sp}$ were taken from ref. 25.

the trend does not persist in other garnets that possess considerably higher damping, e.g., TmIG[45], EuIG[46] or in other FMs. In these garnets, the spin mixing conductivity may be affected by proximity effects which were shown to induce interfacial ferromagnetism[47]. Therefore, additional $\alpha_{sp}$ measurements were carried out for the complete set of NM/Py bilayers. These bilayers consisted of $25NM|10Py$ (units in $nm$), and the $\alpha_{sp}$ measurements were carried out using an optically probed spin-torque ferromagnetic resonance technique[23] (see Supplementary Note 8 for additional details). In the Py-based bilayers, $\alpha^{Py}_{sp}$ is defined as $\alpha_{NM/Py} - \alpha_{Py}$, where $\alpha_{NM/Py}$ is the measured damping of the bilayer and $\alpha_{Py}$ is the intrinsic damping of Py, determined by measuring a bare $10\,nm$ Py film[25]. The extracted $\alpha^{Py}_{sp}$ values are also summarized in Table 2 and are plotted in the inset of Fig. 4f. A good correlation with $\sqrt{\tilde{V}}_{RMS,norm}$ is found having an R-square determination parameter of 0.947. The slight reduction in the correlation as compared to the YIG-based case can be attributed to the higher intrinsic $\alpha$ of Py making it a less ideal for quantifying the NM-induced enhancement of the Gilbert damping. Additionally, in contrast to the YIG-based samples of ref. 25, the Py layer was deposited on top of the NM layer which may have resulted in some variation of the intrinsic damping of Py from sample to sample.

The same measurements were repeated using a red $638\,nm$ laser and are presented in Supplementary Note 9, where a similar noise behavior is observed. As compared to the measurements at $440\,nm$, the $|\Phi_K|$ values at $638\,nm$ are smaller, and the ordering of the magneto-optical responses is also altered. In Cu and Au, for example, this behavior may be related to the features of the energy band and to the smaller contribution of the 3d and 5d interband transitions. The correlation of the noise with $\alpha_{sp}$ persisted also at $638\,nm$ resulting in an R-square determination parameter of 0.915. No correlation with $a^{1/2}_{B_{mod}}$ was observed given the new ordering of the responses.

## Discussion

SOC exists in all materials including FMs and NMs. In FMs it stands behind $\alpha$, whereas in NMs the corresponding quantity is the SOC parameter, $\xi_{NM}$. In thin FM films, it was shown that $\alpha$ is proportional to $\xi^2_{FM}$[48,49], where $\xi_{FM}$ is the SOC parameter of the FM. $\alpha_{sp}$ originates from the dissipation of spin angular momentum in the NM but is expressed in terms of losses of a magnetic system and is fundamentally related to $\xi_{NM}$. The mechanism responsible for the dissipation of spin angular momentum in the NMs relevant to our experiments is the spin relaxation by the Elliott–Yafet mechanism. Within this framework, spin-flip accompany electron scattering events such that the spin relaxation time, $\tau_s$, is related to $\tau$ according to $\tau^{-1}_s \propto \xi^2_{NM} \cdot \tau^{-1}$[50]. This is the same dependence on $\xi_{NM}$ that was found to describe $\alpha_{sp}$ in FM/NM bilayers[51]. It seems that the fluctuation–dissipation theorem leads to a similar conclusion. Accordingly, the power spectral density of the spontaneous fluctuations can be expressed by $S(\omega) = \frac{2k_BT}{\omega}\chi''(\omega)$, where $\chi''(\omega)$ is the dissipative part of the response function, $k_B$ is Boltzmann's constant, $T$ is the temperature, and $\omega$ is the frequency. Therefore, in

magnetic systems, $S(\omega)$ is proportional to $\alpha$. Analogously, it is plausible that in NMs the spin fluctuations would be proportional to $\alpha_{sp}$.

It is well known that the interaction between the optical radiation and spins in solids is mediated through SOC. The early investigation by Argyres[52] considered the relativistic magnetic field that emerges from spin-orbit interaction upon the motion of the electron in the crystal field. In FMs, where the spin-up and spin-down populations are uneven, this strong relativistic magnetic field is responsible for the large Kerr response. In NMs, where the spins are evenly distributed, the Kerr signal originates only from the Lorentz force induced by the external magnetic field which is spin-independent. However, each random spin still experiences a strong, spin-dependent relativistic field that affects its trajectory. This can induce the observed electromagnetic noise, which depends on the strength of the SOC.

Recently, we showed that $\alpha$ plays a pivotal role also in the interaction between the magnetic component of optical radiation and spins in solids[53]. Accordingly, a magnetic optical field that is circularly polarized was shown to exert a torque on magnetic moments. This interaction was described by a strength parameter, $\eta$, that scales with $\alpha$ according to $\eta = \alpha\gamma 2\pi H_{opt}/\omega_{opt}$, where $\gamma$ is the gyromagnetic ratio of the electron and $H_{opt}$ is the amplitude of the helical optical magnetic field. In contrast, linearly polarized light is known to disarrange the spins and reduce the magnetic order parameter as seen in all-optical helicity dependent switching experiments[54–57]. The correlation between $\alpha_{sp}$ and $\sqrt{\tilde{V}}_{RMS,norm}$ suggests that the linearly polarized beam induces spin fluctuations that scale with $\alpha_{sp}$ and enhance the electromagnetic noise.

Hall measurements stand out as a cornerstone technique in material research and solid-state physics and bridge the gap between fundamental research and practical applications. Having overcome the need for physical contacts, the OHE offers numerous advantages over traditional contact-based methods with the primary being its non-invasive and non-destructive nature. At visible wavelengths, the convenience of using laser radiation offers advantages in spatial and temporal resolution, breadth of materials, and temperature and environment flexibility. The relevance of the presented technique to short wavelengths mitigates the influence of carrier scattering and is expected to facilitate the research of band structure physics and topological materials. Future extensions of the work should broaden the spectrum of materials to include additional metals, semiconductors, multi-layered films, and topological and 2D materials. A temperature-dependent measurement is of particular interest, as it could offer key insight into the noise mechanisms and underpin a deeper understanding of their origin. Furthermore, distinguishing between $\theta_k$ and $\epsilon_k$ in non-magnetic materials, for example, by modifying the detection line, is key to resolving the intricate details of the interaction of light with matter. The superior sensitivity of the technique paves the way towards discovery of new phenomena and applications such as an optical determination of the spin-orbit interaction.

## Methods

### Optical setup

In our experiments, $l$ varied between $1 - 10\ mm$ while the probed area was determined from the laser spot size which was $\sim 0.5 \times 0.5\ mm^2$. The magnetic disc was $10\ cm$ in diameter and consisted of 15 magnets each of a diameter of $7.5\ mm$ resulting in $\omega_{mod} = 2\pi \cdot 750\ rad/sec$. The minimal incremental motion of the motorized stage was $0.05\ \mu m$ such that $B_{AC}$ was controllable to a resolution of $\sim 0.5\ \mu T$ at the closer end of the translation stage. The largest applicable $B_{AC}$ was $\sim 0.45\ T$. The "Coherent Cube" model 445-40C laser was used in the experiments and the light modulation was achieved electrically by connecting the reference signal of the lock-in amplifier (Stanford Research SR830) to the "Laser Enable" control input. In the measurement of $V_{PD}^0$, the dynamical range of the photodiode (Thorlabs DET36A2) was extended by additional optical attenuation.

### Sample fabrication

All samples of this work were grown by Ar-based DC magnetron sputtering on 2-inch Si/SiO2 wafers. The substrates were cleaned using the Piranha method to remove any organic residues. The base pressure before deposition was $8 \times 10^{-10}\ Torr$ and $3 \times 10^{-3}\ Torr$ during deposition with sputtering power of $125\ W$. The base pressure was achieved by pre-sputtering Ti for 5 min. The TaN capping layer was grown using a gas mixture of nitrogen (50%) and argon (50%). The electrical DC Hall measurements were carried out on dies of $10 \times 10\ mm^2$ in the Van der Pauw configuration.

### Reporting summary

Further information on research design is available in the Nature Portfolio Reporting Summary linked to this article.

## Data availability

The data that support the findings of this study are available from the corresponding author upon reasonable request.

## Code availability

The codes used in theoretical calculations are available from the corresponding author upon reasonable request.

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

## Acknowledgements
This work was supported by the Israel Science Foundation (Grant No. 3011/23 and 1217/21), and the Cooperation in Science and Technology action (CA23136 CHIROMAG). N.A.S., N.B., B.A., and A.C. thank the Harvey M. Krueger family center of Nanoscience and Nanotechnology at the Hebrew University of Jerusalem and the Lipper and Peter Brojde foundations. N.A.S and A.C. thank Dr. Dror Shafir and Dr. Yael Kurzweil Segev for fruitful discussions.

## Author contributions
N.A.S. and A.C. conceptualized the research. A.C. supervised the experimental work. N.A.S. was responsible for sample fabrication and characterization. The measurement system was developed by N.A.S., N.B., and A.R., while N.A.S., A.R., and M.M. carried out the measurements. The theoretical framework was developed by N.A.S., A.C., B.Y., I.R., D.K., and T.H. and B.A. contributed to the discussion of the spin-orbit coupling correlation. All authors participated in discussing the results and contributed to writing the manuscript. The research was directed by A.C.

## Competing interests
The authors declare no competing interests.
