## [Peer Review File · Nature Communications]

A sensitive MOKE and optical Hall effect technique at visible wavelengths: insights into the Gilbert damping

Corresponding Author: Professor Amir Capua

Version 0:

Reviewer comments:

Reviewer #1

(Remarks to the Author)

The manuscript "A sensitive MOKE and optical Hall effect technique at visible wavelengths: insights into the Gilbert damping", by Am-Shalom et al., describes a method for magnetic field modulation by circular motion of permanent magnets placed on a disk, this field modulation has been reported in a previous work by Rothschild et al. With this set up, the authors investigate the optical Hall effect (OHE) on several transition metals, and attempt to evaluate theoretically the observed trend in their measurements.

In my opinion, this work, although interesting in its own way, is not appropriate for Nature Communications for the following reasons:

1. Novelty of experimental technique: although the described technique is quite ingenious, it has been reported in a previous manuscript of the same group (Ref. [27]).

Moreover, I believe the experimental apparatus can be furthermore improved to provide meaningful results. The authors only work with the measured voltage from their lock-in amplifier which does not provide by itself an appropriate way for comparing the samples, as it appears the authors did not consider factors such as sample reflectance/absorptance. The authors should process their data in a proper way to report an intensive property of the material.

The schematic also shows that the reflected probe beam goes through another polarizer, which would cause that the probe beam in the detection line would be neglecting the added ellipticity by the samples, only measuring the cosine component of the Kerr rotation. The authors should consider improving the detection line as described in <https://doi.org/10.1103/PhysRevB.106.014410>, to separate the real/imaginary parts from the change in Kerr angle.

2. Sample characterization: the authors do resistivity measurements on the metallic film samples, in order to use Drude model to fit the measured voltage. How does the measured resistivity compares with previously reported values for those materials?

In MOKE experiments from other authors, it has been observed that for tantalum the phase will cause a variation in its optical response (<https://doi.org/10.1103/PhysRevMaterials.7.125202>). For this matter, the authors should consider extending the characterization of the materials (morphology, structure, etc.) in order to establish any possible correlation between the measured values and these properties.

3. Damping: the authors attempt to correlate the supposed damping on the metallic layers with the measurements obtained. They use the manuscript on Ref[28] to extract the parameters for Gilbert damping. This approach has several problems: first, Ref[28] measures the dynamic magnetic properties of YIG/metal interfaces, hence the reported values correspond to these interfaces only. For other garnets, the Gilbert damping is considerable larger (<https://doi.org/10.1103/PhysRevB.94.140403>, <https://doi.org/10.1103/PhysRevMaterials.5.124414>), therefore it is unknown if the trend observed in YIG/metal will be the same as on TmIG or EuIG, for example; therefore we cannot make any conclusions about the sole metal layer itself based on Ref[28].

Reviewer #2

(Remarks to the Author)

Review report for "A sensitive MOKE and optical Hall effect technique at visible wavelengths: insights into the Gilbert damping"

The authors of Am-Shalom et al. measured the magneto-optical Kerr effect (MOKE) on non-magnetic metals of Cu, Au, Ta, and Pt with application of external magnetic field. To enhance the sensitivity of MOKE, the authors modulated the magnetic field using permanent magnets that are placed on a rotating disc. The magnitudes of MOKE results were compared with the Lorenz-Drude theory. In addition, the noise floors of MOKE results were compared with the damping constant. Their method and interpretation are interesting. However, before the physical interpretation, I need to check the reliability of their experiment itself. Although I am familiar with the MOKE experiment, their way of data representation is somewhat strange and hard to understand for me. I cannot recommend this manuscript in the present form.

1. The authors presented their results as the photodiode voltage (VPD). But the conventional way of the MOKE representation is the Kerr rotation (θ_K), the polarization rotation of the reflected light. Please show the explicit relation between VPD and θ_K .
2. The data show that VPD is quadratically proportional to the applied magnetic field (BDC or BAC of Figs. 2, 3, and 4). I cannot understand this relation. The Kerr rotation should be linearly proportional to the induced magnetization (ΔM), which is linearly proportional to the applied magnetic field. In addition, I expect that the Kerr rotation and the photodiode voltage has a linear relationship. Then, in my understanding, VPD should be linearly proportional to the applied magnetic field.
3. In Figs. 2 and 3, the authors compared the B-field modulated signal and light-modulated signal. For the light-modulated signal, what is the modulation device? Is it a photo-elastic modulator? What is the modulation frequency? The applied field is not modulated, so the light-modulated signal was achieved with BDC not BAC, am I right? Please provide a more detailed information for the light-modulated signal.

Version 1:

Reviewer comments:

Reviewer #1

(Remarks to the Author)

After reviewing the revised manuscript by Am-Shalom et al., I consider that the authors have addressed the comments posed in the first round of reviews. However, I found that by adding the experiment on Py/NM samples to back up the results presented in the first draft, when the text switches to the YIG section, the transition seems awkward. I could not find any additional information on the YIG samples. I believe that it is quite important that the authors present the data regarding the YIG films they used in their experiments (such as grow method, substrate, thickness, magnetic properties). This is important because now that the authors added the (more complete) experimental data on the permalloy samples (such as the OSTFMR), I consider it is quite important that the YIG samples receive at least the same degree of detail as the permalloy/NM samples.

Additional comments:

1. In the SI, the XRD presented omits the substrate on which the measured films were grown, this information should be reported.

Reviewer #2

(Remarks to the Author)

The response letter and revised manuscript do not satisfy my major concern. Their main data were the photodiode voltage response that are quadratic to the applied B-field. Although the equation shows the quadratic dependence on the B-field at the analyzer angle of 90 degree, it is still questionable about the origin of this signal. I expect that other mechanisms, such as magnetostriction might cause the B-field-quadratic signal. The cross-polarizer setup is widely used in the MOKE community, but almost always the linear response regime is used. Even in a 2D ferromagnet, which has a much lower MOKE response compared to 3D ferromagnets, a linear response, rather than a quadratic response, was used by intentionally rotating the analyzer angle from 90 degree (see Supplementary Section 2 of Nature Nanotechnology, 19, 1290 (2020)). To support authors' drastic claim of the correlation between the noise level and spin-orbit coupling, I suggest a re-examination of their data in the linear response regime. Because of this concern for the credibility issue, I do not recommend a publication.

Version 2:

Reviewer comments:

Reviewer #2

(Remarks to the Author)

I raised a concern about the validity of the quadratic relationship between the photodiode voltage and the B-field. The

authors measured the photodiode voltage of the permalloy films at different deviation angles from the cross-polarization arrangement. In Fig. S2 (c), the measured photodiode voltage can be linear to the B-field at the deviation angle of 0.06 degree. At the deviation angle of 0 degree, the photodiode voltage is linear to the B-field at the low-field region and becomes nonlinear at the high-field region. However, this behavior cannot be described by a pure quadratic term. In my opinion, this result suggests that the pure quadratic response is almost impossible to achieve. Therefore, I cannot accept the relation between the square root of the photodiode voltage and the Kerr rotation. Considering this experimental concern and lack of theoretical analysis for the noise relation to spin-orbit coupling, I do not recommend the publication of this manuscript in this journal.

Reviewer #3

(Remarks to the Author)

Referee Report on the manuscript entitled "A sensitive MOKE and optical Hall effect technique at visible wavelengths: insights into the Gilbert damping" by Amir Capua et al., submitted to Nature Communications.

I have read the revised manuscript carefully, including all Supplement Notes, and the latest second-round rebuttal. I appreciate the authors' great effort to improve the manuscript by providing detailed explanations of the reviewers' concerns, which were added in the text of the manuscript and in additional Supplement Notes. However, I share the concerns of Referee #2, for the reasons given below, which make my opinion negative about the suitability of the manuscript for publication in Nature Communications in its current form.

I will divide my comments into Part A about the experimental results, and Part B about the interpretation.

A1. The authors determine the Kerr rotation values FiK as a function of the magnetic field B for a laser wavelength of $\lambda=440$ nm. They compare the measured FiK values for Cu, Au, Ta and Pt (Fig.4c) with those obtained for Py (Fig.2b). However, there is no basic comparison of the FiK values with the ones reported in the literature for the metals studied. For $\lambda=440$ nm, i.e. the photon energy ~ 2.8 eV, the measured FiK value in Ref. [20] is ~ 0.4 mdeg for Cu and ~ -0.4 mdeg for Au in an alternating magnetic field B of amplitude 1 T. Exactly corresponding values (0.6 mdeg and -0.6 mdeg, respectively) were measured in Ref. [10] in a constant field $B=1.5$ T. However, the FiK values measured in the manuscript for these metals are more than an order of magnitude higher (~ 3 mdeg at $B=0.45$ T for Cu and Au, Fig.4(c)) than the reported literature data!

A2. The fundamental discrepancy between the measured large FiK values and the literature data strengthens concerns, already expressed by Referee #2, about the origin of the measured signal, that a mechanism other than Kerr rotation may be responsible for the measured FiK . To resolve this discrepancy, my recommendation is that the authors should measure and compare the Kerr rotation on the Al film, for which the FiK at 0.45 T, estimated from the data measured and reported in the literature for Al metal (Ref. [19]), is smaller (0.18 mdeg for $B=0.45$ T), and the plasma frequency is by $\sim 50\%$ higher than for Cu and Au.

B1. In line 175 of the revised manuscript, the following sentence is written "Core-level transitions are neglected as they occur in the X-ray range". This statement is true, but the authors forgot that 3d and 5d interband transitions cannot be neglected. As it is well known, at such a high photon energy as 2.8 eV, the interband transitions are dominant and masking the free-electron-contribution intraband transitions (see, e.g., Refs. [20] and [10]).

B2. Why did the authors choose a blue laser as the light source? Using a red or infrared laser would allow measurement below the energy edge of interband transitions, i.e., only the free-electron contribution in the studied metals.

In conclusion, the last statement in the abstract of the manuscript "These results suggest that the electromagnetic noise arises from optical interactions with the spins and highlight a possible avenue for measuring the spin-orbit interaction using optical techniques" requires, in my opinion, more justification, both experimental and theoretical. Additional test measurements on Al reference sample, for which Kerr rotation is small, plasma frequency is high, and spin-orbit interaction is from one to two orders of magnitude smaller than for the metals studied, are highly recommended.

Reviewer #4

(Remarks to the Author)

Please see attached pdf.

Version 3:

Reviewer comments:

Reviewer #3

(Remarks to the Author)

The authors have adequately addressed my concerns and I find the revised manuscript satisfactory. In my opinion, the manuscript in the present form is suitable for publication in Nature Communications.

Reviewer #4

(Remarks to the Author)

Authors modified the manuscript and addressed the concerns according to my previous review. I think that they modified the manuscript properly and the manuscript is now acceptable to be published.

As the manuscript was already reviewed by other referees, one of the issues given by the other referee is validity of quadratic dependence of MOKE signals with respect to magnetic field. I have read authors rebuttal and found that such quadratic dependence is natural in the perfect cross-polarizer configuration and also reported in the previous study [Li *et al.* Appl. Phys. Lett. **124**, 054001 (2024)] as authors mentioned. The analyzer angle was slightly rotated to detect linear magnetic response in the most of previous works. However, I also found a thing different with the point mentioned by the other referee that the measurement technique reported in the present manuscript is different with the previous study, which might cause the confusion. Authors used lock-in amplifier to detect modulated MOKE signals that is proportional to square of Kerr rotation or magnetic field. That is,

$$r_{ps} \propto \sin(\omega_{\text{mod}}t), r_{ps}^2 \propto \sin^2(\omega_{\text{mod}}t) = \frac{1}{2}(1 + \cos(2\omega_{\text{mod}}t))$$

So, the signal measured at the photo-diode is proportional to the signal with the frequency twice larger than the modulated signal and the dc signal. I feel that this fact should be clarified in the revised manuscript.

Another concern about the work I think is that the physical mechanisms of the correlation between noise level and spin-orbit coupling in nonmagnetic metals are not discussed in detail in the present manuscript. For example, Gilbert damping parameter and thermal fluctuation of magnetization are related due to the fluctuation-dissipation theorem. I think that the noise in nonmagnet may be caused by the following mechanism; the spin induced by magnetic field (Pauli susceptibility) in nonmagnet has particular life-time, namely spin relaxation time which should be related to spin-orbit coupling of the material according to Elliot-Yafet mechanism. Then, this dissipation is equal to thermal fluctuation of spin. Authors should describe the mechanism of noise observed in nonmagnetic metals in the revised manuscript. I think that discussions in the present manuscript rely on the Gilbert damping which is the parameter for ferromagnet. However, authors measure the noise without ferromagnet, which contradicts the discussion presented. In addition to the revision of discussions, temperature dependent measurement might provide more insights into the mechanism if possible.

Response to comments made by Reviewer #1

“1. Novelty of experimental technique: although the described technique is quite ingenious, it has been reported in a previous manuscript of the same group (Ref. [27]).”

Although the two methods utilize a magnetic field modulator they are fundamentally different. Ref. [27] describes a sensitive ferromagnetic resonance (FMR) technique whereas the technique at hand is an optical Hall/MOKE method. The technique of Ref. [27] is purely electrical, requires a microwave source, an electrical RF waveguide, and measures the microwave absorption. Here, the technique is purely optical, has none of the key components above, and rather than probing the magnetization precessions at the GHz frequencies, the optical response is measured. The only element the two share in common is the use of the magnetic field modulator to enhance the sensitivity.

To avoid possible confusion, we have added additional explanations that will assist the readership to distinguish between the two quite different methods.

“Moreover, I believe the experimental apparatus can be furthermore improved to provide meaningful results. The authors only work with the measured voltage from their lock-in amplifier which does not provide by itself an appropriate way for comparing the samples, as it appears the authors did not consider factors such as sample reflectance/absorptance. The authors should process their data in a proper way to report an intensive property of the material.”

The revised manuscript now presents the data also in terms of the magnitude of the complex Kerr angle, $|\Phi_K|$. $|\Phi_K|$ indicates the strength of the MOKE and is useful for quantitative comparison to other works and materials. To this end, the reflectance/absorptance was measured as well.

Φ_K is given by $\Phi_K = \theta_k + i\epsilon_k$, where θ_k is the Kerr rotation and ϵ_k is the ellipticity. In the maximal extinction cross-polarized arrangement of the setup, only the magnitude of Φ_K can be measured. Using the Jones calculus and following the derivation by Gomez, Wilson et. al. [Rev. Sci. Instr. 91 023905 (2020)], it can be shown that for an incident s -polarized field, $\Phi_K = \frac{r_{ps}}{r_{ss}}$, where r_{ss} is the Fresnel reflection coefficient of the s -polarization and r_{ps} is the MOKE reflection coefficient responsible for the transverse polarization. In the cross-polarized detection line: $V_{PD} \propto |E_0^i|^2 \cdot |r_{ps}|^2$, where E_0^i is the amplitude of the incident laser field. Since $|r_{ps}| \ll |r_{ss}|$, the lock-in amplifier voltage, V_{PD}^0 , corresponding to $|r_{ss}|$ can be measured by placing the photodiode immediately after the sample. Accordingly, $|\Phi_K| = \sqrt{V_{PD}}/\sqrt{V_{PD}^0}$. For the Py sample, $|\Phi_K|_{@450\text{ mT}} = 40\text{ mdeg}$, which is comparable to values reported in the literature.

In the original version of the manuscript, the different reflectance/absorptance of each sample were accounted for in the model derived for V_{PD} . The model for V_{PD} takes into account the different Fresnel coefficients of the samples as well as the absorption of the sample. These effects eventually led to the appearance of the factor $A(n)$ in the expression for V_{PD} (Eq. (2) in the revised manuscript), where n is the complex index of refraction.

Following the comment made by the reviewer, in the revised manuscript we have added a new paragraph where the extraction of $|\Phi_K|$ is described and is based on new measurements of the reflectance/absorptance coefficients. $|r_{ss}|$ of each sample appears now in ‘Methods’. Figure 2(b) was updated and a new Fig. 4(c) was added to present $|\Phi_K|$ data. Additionally, a new detailed Supplementary Note 1 was included where the relevant mathematical derivation is presented. The Lorentz-Drude model was extended to also include the expression for $|\Phi_K|$. The main text now elaborates on the meaning of the factor $A(n)$.

“The schematic also shows that the reflected probe beam goes through another polarizer, which would cause that the probe beam in the detection line would be neglecting the added ellipticity by the samples, only measuring the cosine component of the Kerr rotation. The authors should consider improving the detection line as described in <https://doi.org/10.1103/PhysRevB.106.014410>, to separate the real/imaginary parts from the change in Kerr angle.”

Our work relies on enhancing the sensitivity by applying B -modulation which enables to measure the weak $|r_{ps}|^2$ term stemming purely from the MOKE. It is not clear that the B -modulation scheme will have added value when integrating the detection line described by Ortiz et. al. in [PRB **106**, 014410 (2022)].

The detection line described by Ortiz et. al. is based on a balanced detector where the generated photocurrent at each port results from both the s and p components of the polarization:

$$\begin{bmatrix} V_1 \\ V_2 \end{bmatrix} \propto |E_0^i|^2 \begin{bmatrix} |r_{ss}|^2 + r_{ss}r_{ps}^* \exp(i\gamma) + r_{ss}^*r_{ps} \exp(-i\gamma) + |r_{ps}|^2 \\ |r_{ss}|^2 - r_{ss}r_{ps}^* \exp(i\gamma) - r_{ss}^*r_{ps} \exp(-i\gamma) + |r_{ps}|^2 \end{bmatrix}.$$

Using a controllable retarder to vary the retardation γ , θ_k and ϵ_k are then extracted from the difference signal $V_1 - V_2$:

$$V_{diff} \propto |E_0^i|^2 \cdot \text{Re}\{r_{ss}r_{ps}^* \exp(i\gamma)\}.$$

It is seen that V_{diff} eliminates the purely MOKE signal $|r_{ps}|^2$ independent of the type of modulation. Moreover, from the expression for V_{diff} , it is seen that B -modulation is

not necessarily advantageous over light-modulation scheme since both modulate V_{diff} in the same fashion: The two quadratic terms were already eliminated when taking the difference $V_1 - V_2$. Additionally, the detection of $|r_{ss}|^2$ in the balanced detection scheme may introduce additional background shot-noise that may potentially limit the measurement of non-magnetic metals.

In the revised manuscript we have added a discussion and further explanations related to the integration of the detection line described by Ortiz et. al. These appear in the main text as well as in a separate Supplementary Note 3 and in the summarizing paragraph. Extraction of θ_k and ϵ_k is beyond the scope of the present manuscript and is left for future work.

“2. Sample characterization: the authors do resistivity measurements on the metallic film samples, in order to use Drude model to fit the measured voltage. How does the measured resistivity compares with previously reported values for those materials? In MOKE experiments from other authors, it has been observed that for tantalum the phase will cause a variation in its optical response (<https://doi.org/10.1103/PhysRevMaterials.7.125202>). For this matter, the authors should consider extending the characterization of the materials (morphology, structure, etc.) in order to establish any possible correlation between the measured values and these properties.”

The resistivities extracted from the DC Hall transport measurements are summarized in the table below. For comparison, the table also includes values reported in the literature. The measured resistivities are consistent with previously reported values. The resistivity of Ta corresponds to the β -phase as expected for the growth conditions. Resistivity values have been added to the main text and Supplementary Note 4.

Material	Hall measurements [$\mu\Omega \cdot cm$]	Ref. 1 [$\mu\Omega \cdot cm$]	Ref. 2 [$\mu\Omega \cdot cm$]
Pt	19.2	20 (J. Appl. Phys., 122 , 025107, 2017)	20.5 (Appl. Surface Science, 254 , 7356, 2008)
Ta	173	172 (Acta Materialia, 183 , 504, 2020)	170 (Thin Solid Films, 688 , 137403, 2019)
Au	3.9	5.8 (Phys. Rev. B., 70 , 165414, 2004)	3.4 (J. Phys. F: Met. Phys., 18 , 331, 1988)
Cu	3.3	2.75 (J. Appl. Phys., 120 , 065106, 2016)	3.09 (Int. J. of Nano Dimension 3, 217, 2013)

Resistivity extracted from DC Hall transport measurements.

X-ray diffraction (XRD) measurements were carried out on all samples. The data is presented below and confirms that the Ta film was grown in the β -phase and that the Pt, Au, and Cu samples grew in the preferred (111) crystal orientation. The extracted lattice parameters for all crystals agree with tabulated data. XRD data is now described in the main text and is presented in Supplementary Note 4.

XRD characterization of the 50 nm films of (a) Pt. (b) Ta. (c) Au. and (d) Cu. Arrows indicates the (111) crystal orientation in the Pt, Au, and Cu spectra. Arrows in Ta data corresponds to the β phase of Ta.

The surface morphology was characterized using atomic force microscopy (AFM). The RMS surface roughness values, R_q , of the samples were $R_{q,Pt} = 0.384$ nm, $R_{q,Ta} = 0.27$ nm, $R_{q,Au} = 0.55$ nm, and $R_{q,Cu} = 0.71$ nm. The figure below presents the measured AFM results. AFM data is also included in Supplementary Note 4.

AFM measurements of (a) Pt. (b) Ta. (c) Au. and (d) Cu.

We could not identify any explicit correlation between the AFM and XRD data and the optical measurements. Nevertheless, the additional characterization data confirms the proper film quality.

“3. Damping: the authors attempt to correlate the supposed damping on the metallic layers with the measurements obtained. They use the manuscript on Ref [28] to extract the parameters for Gilbert damping. This approach has several problems: first, Ref[28] measures the dynamic magnetic properties of YIG/metal interfaces, hence the reported values correspond to these interfaces only. For other garnets, the Gilbert damping is considerable larger (<https://doi.org/10.1103/PhysRevB.94.140403>, <https://doi.org/10.1103/PhysRevMaterials.5.124414>), therefore it is unknown if the trend observed in YIG/metal will be the same as on TmIG or EuIG, for example; therefore we cannot make any conclusions about the sole metal layer itself based on Ref [28].”

Generally, the different interfaces can influence the dynamical measurements by proximity-induced ferromagnetism in the NM and by a spin-pumping effect as discussed in garnets by Chi et. al [Phys. Rev. B 94, 140403(R)]. To address the Reviewer’s concern, we have carried out additional Gilbert damping (α) enhancement measurements for the complete set of Py/NM bilayers. Py has higher α as compared to YIG, that is closer to the high α of TmIG or EuIG. The α measurements were carried out using an optically-probed FMR technique and the α enhancement factor, α_{sp} , was extracted in the usual manner by measuring an additional bare Py film such that $\alpha_{sp}^{Py} = \alpha_{NM/Py} - \alpha_{Py}$. Here, $\alpha_{NM/Py}$ is the α of the bilayer structure and α_{Py} corresponds to the damping of the bare Py film. The results are summarized in the figure below. The R-square determination parameter in this case is 0.95 and is also quite high illustrating that the correlation persists also in the high α case. For comparison, in the YIG-based bilayers of Ref. [28], this parameter was 0.99. The higher correlation observed in the YIG-based bilayers may be related to the lower intrinsic damping of YIG and the more

Correlation between the noise α enhancement in Py-based bilayers.

substantial contribution to the losses of angular momentum stemming from the NM. Additionally, in Ref. [28], the YIG layer was deposited directly on the substrate while the NMs were grown on top, therefore, the intrinsic damping (α_{Py}) was similar in all samples. In our samples, the Py was grown on top of the NM in order to probe the dynamics optically. Therefore, strain effects and lattice mismatch may have led to some variation of α_{Py} .

In the revised version of the manuscript, we have added the discussion of the α enhancement in the case of higher α that is closer to the values of TmIG and EuIG. The Py-based damping measurements are presented now in Fig. 4(f) together with the values measured in the YIG-based bilayers. The procedure of measuring the damping values is described in detail in the new Supplementary Note section 5. This note includes a detailed description of the optically-probed FMR method, fabrication of the new samples, data analysis method, and the measured responses.

- We thank the Reviewer for their comment and for the careful examination of our manuscript.

Response to comments made by Reviewer #2

“1. The authors presented their results as the photodiode voltage (V_{PD}). But the conventional way of the MOKE representation is the Kerr rotation (θ_K), the polarization rotation of the reflected light. Please show the explicit relation between V_{PD} and θ_K .”

In the revised manuscript we present the data in a more conventional manner that is useful for quantitative comparison with other reports and materials. We represent the data in terms of the magnitude of the complex Kerr rotation, $\Phi_K = \theta_k + i\epsilon_k$, which indicates the strength of the MOKE. In the expression for Φ_K , θ_k is the Kerr rotation and ϵ_k is the ellipticity. V_{PD} measures the transverse polarization which originates from both θ_k and ϵ_k . In the cross-polarized polarizer-analyzer arrangement, it is impossible to distinguish between the two, but rather, their geometrical average can be measured: $|\Phi_K| = \sqrt{\theta_k^2 + \epsilon_k^2}$.

To determine the relationship between $|\Phi_K|$ and V_{PD} we use the Jones calculus. For the s -polarized incident beam of our experiments and an analyzer that transmits the p -polarization, it can be shown that $|\Phi_K| = \frac{|r_{ps}|}{|r_{ss}|}$, where r_{ss} is simply the Fresnel coefficient for the s -polarization and r_{ps} represents the MOKE reflection coefficient. In the cross-polarized arrangement, $V_{PD} \propto |E_0^i|^2 \cdot |r_{ps}|^2$, where E_0^i is the amplitude of the incident laser field. Since $|r_{ps}| \ll |r_{ss}|$, $|r_{ss}|$ was measured by placing the photodiode immediately after the sample, and recording the lock-in voltage, V_{PD}^0 . Accordingly, $|\Phi_K| = \sqrt{V_{PD}} / \sqrt{V_{PD}^0}$ was determined.

For Py, we measured $|\Phi_K|_{@450\text{ mT}} = 40\text{ mdeg}$, which is comparable to values measured in similar samples. For the metallic films we measured $|\Phi_K|_{@450\text{ mT}} = 4.77, 3.64, 3.39, \text{ and } 3.08\text{ mdeg}$ for Pt, Ta, Au, and Cu, respectively.

In the revised manuscript an additional paragraph was added to describe the procedure of extraction of $|\Phi_K|$. Figures 2(b) was updated and Fig. 4(c) was added to present $|\Phi_K|$ data. Additionally, a new Supplementary Note 1 was added and presents the mathematical derivation in detail.

“2. The data shows that V_{PD} is quadratically proportional to the applied magnetic field (B_{DC} or B_{AC} of Figs. 2, 3, and 4). I cannot understand this relation. The Kerr rotation should be linearly proportional to the induced magnetization (ΔM), which is linearly proportional to the applied magnetic field. In addition, I expect that the Kerr rotation and the photodiode voltage has a linear relationship. Then, in my understanding, V_{PD} should be linearly proportional to the applied magnetic field.”

The relationship between the measured voltage, V_{PD} , and the applied magnetic field, depends on a variety of experimental parameters including:

- 1) The precise polarizer-analyzer cross-polarization angle.
- 2) The type of modulation applied, e.g. light-modulation (L_{mod}), B -modulation (B_{mod}), or polarization modulation by a photoelastic modulator (PEM).
- 3) The type of photodetector used in the detection line, i.e. a single port or a balanced photodetector.

For example, slightest deviations from the ideal cross-polarized polarizer-analyzer arrangement will immediately result in the familiar linear relationship mentioned by the Reviewer. Below we explain the influence of these parameters both theoretically and by additional experiments. For simplicity, the applied magnetic field is indicated here by B , rather than B_{DC} or B_{AC} .

To determine the relationship between V_{PD} and B , we use once more the Jones calculus. V_{PD} depends on orientation of the analyzer. The Figure below illustrates schematically the optical arrangement. For the s -polarized incident beam and an analyzer having a transmission axis oriented at an angle ϕ relative to plane of incidence, it can be shown that:

$$V_{PD} \propto |E_0^i|^2 \cdot \left[\sin^2 \phi |r_{ss}|^2 + \cos^2 \phi |r_{ps}|^2 + 2 \cos \phi \sin \phi \cdot \text{Re}\{r_{ss}r_{ps}^*\} \right]. \quad (1)$$

In Eq. (1), the first term is independent of B and the second term stems purely from the Kerr effect. As the Reviewer indicated, $|r_{ps}| \propto B$, therefore, the second term scales with B^2 . The third term depends linearly on B and stems from an intermixing between the primary s -polarized beam and the Kerr response.

In the perfect cross-polarization arrangement, $\phi = 0^\circ$, and $V_{PD} \propto |E_0^i|^2 \cdot |r_{ps}|^2$. Therefore, $V_{PD} \propto B^2$ as seen in the experiments. However, since $|r_{ss}|^2 \gg |r_{ps}|^2$, for

Schematic illustration of the experimental setup.

very small nonvanishing ϕ , the first and third terms can dominate and modify the relationship between V_{PD} and B .

V_{PD} is the demodulated signal measured by the lock-in amplifier and depends on the applied modulation scheme. The two modulation cases, B_{mod} and L_{mod} , appearing in the manuscript are described below:

1) B modulation:

In the case of B_{mod} , only the second and third terms of Eq. (1) will be measured by the lock-in amplifier such that $V_{PD,B_{mod}} \propto |E_0^i|^2 \cdot [\cos^2 \phi |r_{ps}|^2 + 2 \cos \phi \sin \phi \cdot Re\{r_{ss} r_{ps}^*\}]$. The calculated $V_{PD,B_{mod}}$ for small ϕ angles in the range $0^\circ - 0.06^\circ$ are presented in the figure below (panel (a)). As ϕ increases, $V_{PD,B_{mod}}$ becomes linear with B . This is readily seen in panel (b) by normalizing the traces to unity. Already for $\phi = 0.06^\circ$ the linear relationship appears. This behavior was also confirmed experimentally. Panel (c) presents the measured ϕ -dependent V_{PD} for Py and panel (d) presents the same data in normalized units. The measured data agrees well with the calculations confirming that the transformation to a linear dependence occurs at very small deviations.

Calculated and measured V_{PD} for B -modulation and various small ϕ angles. (a) Calculated $V_{PD,B_{mod}}$ for $\phi = 0^\circ, 0.01^\circ, 0.02^\circ$, and 0.06° . (b) Calculated $V_{PD,B_{mod}}$ presented in normalized units. (c) Measured $V_{PD,B_{mod}}$ in Py for $\phi = 0^\circ, 0.01^\circ, 0.02^\circ$, and 0.06° . (d) Measured $V_{PD,B_{mod}}$ in normalized units.

2) Light modulation:

In this case, $|E_0^i|^2$ is modulated. Therefore, the measured lock-in voltage V_{PD} accounts for all terms of Eq. (1):

$$V_{PD,L_{mod}} \propto |E_0^i|^2 \cdot \left[\sin^2 \phi |r_{ss}|^2 + \cos^2 \phi |r_{ps}|^2 + 2 \cos \phi \sin \phi \cdot \text{Re}\{r_{ss}r_{ps}^*\} \right].$$

The additional term $\sin^2 \phi |r_{ss}|^2$ adds a constant B -independent shift to V_{PD} which increases rapidly with ϕ due to the large magnitude of $|r_{ss}|^2$. The figure below presents the calculated and measured ϕ -dependent $V_{PD,L_{mod}}$. The shift is readily seen in the calculated responses in panel (a) while the normalized responses of panel (b) illustrate once more the transition from a parabolic to a linear dependence on B . The corresponding measurements are presented in panel (c) and reproduce the shift in $V_{PD,L_{mod}}$. The normalized measured responses of panel (d) reproduce once more the calculations. Interestingly, it is seen that the L_{mod} measurements are slightly noisier as compared to the measured B_{mod} traces. The small ϕ angles in the experiment were achieved using a standard precision rotational mount.

Calculated and measured V_{PD} for L -modulation and various small ϕ angles. (a) Calculated $V_{PD,L_{mod}}$ for $\phi = 0^\circ, 0.01^\circ, 0.02^\circ$, and 0.06° . (b) Calculated $V_{PD,L_{mod}}$ presented in normalized units. (c) Measured $V_{PD,L_{mod}}$ in Py for $\phi = 0^\circ, 0.01^\circ, 0.02^\circ$, and 0.06° . (d) Measured $V_{PD,L_{mod}}$ presented in normalized units.

It is possible that the high sensitivity of the B -modulation scheme is advantageous in the alignment procedure to achieve a fairly close $\phi = 0$. For comparison, in the L_{mod} scheme, V_{PD} increases significantly with ϕ making it more difficult to detect the weak $|r_{ps}|^2$ term.

Lastly, another common implementation of the detection line utilizes a balanced detector. In this implementation, the reflected beam is rotated by 45° and is passed through a Wollaston prism which splits the beam to the two ports of the balanced detector. The voltage on at each port results from both the s and p components and is given by:

$$\begin{bmatrix} V_1 \\ V_2 \end{bmatrix} \propto \frac{1}{2} |E_0^i|^2 \begin{bmatrix} |r_{ss}|^2 + |r_{ps}|^2 + 2Re\{r_{ss}r_{ps}^*\} \\ |r_{ss}|^2 + |r_{ps}|^2 - 2Re\{r_{ss}r_{ps}^*\} \end{bmatrix}.$$

The MOKE response is then extracted from the difference signal $V_1 - V_2$:

$$V_{diff} \propto |E_0^i|^2 \cdot Re\{r_{ss}r_{ps}^*\}.$$

It is readily seen the $V_{diff} \propto r_{sp}$ which is linear in B . This detection scheme inherently eliminates the purely parabolic MOKE signal $|r_{ps}|^2$ always resulting in a linear relationship with B .

The main text of the revised manuscript includes now a detailed paragraph explaining the possible relationships between V_{PD} and B . Additionally, two new Supplementary Notes (sections 2 and 3) have been added and elaborate on this point. These sections include the detailed mathematical derivation and the measurements.

“3. In Figs. 2 and 3, the authors compared the B -field modulated signal and light-modulated signal. For the light-modulated signal, what is the modulation device? Is it a photo-elastic modulator? What is the modulation frequency? The applied field is not modulated, so the light-modulated signal was achieved with B_{DC} not B_{AC} , am I right? Please provide a more detailed information for the light-modulated signal.”

The light-modulation was achieved electrically by modulating the laser drive current. The laser used in the work was a “Coherent Cube” model 445-40C and the modulation was achieved using the reference signal of the lock-in amplifier that was connected to the “Laser Enable” control input. The bandwidth of this port was 130 KHz and the rise

time was better than $1 \mu\text{sec}$ that was well suited for the 750 Hz modulation rate. The same modulation rate was applied also in the B -modulation case.

In the light-modulation case, the signal was generated using B_{DC} and not B_{AC} . The static magnetic field B_{DC} was applied by fixing the same rotating disc that generated B_{AC} at the maximal field of one of its magnets. B_{DC} was varied by scanning over the sample-magnet distance, l , using the translation stage. Overall, the exact same hardware was used in the light and magnetic field modulation variants.

The manuscript has been revised and the main text includes the details of the application of B_{DC} while the details of the laser and the light-modulation scheme are described in the 'Methods' section.

- We thank the Reviewer for the careful examination of our work and for their comments.

Response to comments made by Reviewer #1

1) *“After reviewing the revised manuscript by Am-Shalom et al., I consider that the authors have addressed the comments posed in the first round of reviews. However, I found that by adding the experiment on Py/NM samples to back up the results presented in the first draft, when the text switches to the YIG section, the transition seems awkward. I could not find any additional information on the YIG samples. I believe that it is quite important that the authors present the data regarding the YIG films they used in their experiments (such as grow method, substrate, thickness, magnetic properties). This is important because now that the authors added the (more complete) experimental data on the permalloy samples (such as the OSTFMR), I consider it is quite important that the YIG samples receive at least the same degree of detail as the permalloy/NM samples.”*

Data of the YIG samples appears now in the main text of the revised manuscript. Additionally, the text was revised such that the transition to the YIG samples is more coherent.

2) *“Additional comments:*

In the SI, the XRD presented omits the substrate on which the measured films were grown, this information should be reported.”

Additional XRD measurements of the bare SI substrate were carried out and appear below:

XRD data of bare Si substrate

These peaks and the contribution of the substrate are now marked in the XRD data of all films. Additional data of the substrates used in work was added to ‘Supplementary Information Note 4’.

We thank the referee for their comments.

Response to comments made by Reviewer #2

1) The response letter and revised manuscript do not satisfy my major concern. Their main data were the photodiode voltage response that are quadratic to the applied B-field. Although the equation shows the quadratic dependence on the B-field at the analyzer angle of 90 degree, it is still questionable about the origin of this signal. I expect that other mechanisms, such as magnetostriction might cause the B-field-quadratic signal.

The quadratic term is not unique to our experiments; others have measured it as well.

The model describing the experiment is well established and the calculation is straight forward. It has been used in a variety of different reports by many different research groups, e.g. J. Appl. Phys. 84 541–546 (1998), Rev. Sci. Instrum. 79, 055107 (2008), Rev. Sci. Instrum. 91, 023905 (2020), J. Appl. Phys. 135, 063901 (2024), Appl. Phys. Lett. 124, 054001 (2024), and many more.

In our previous correspondence, we showed that the measurements agree very well with the model: The responses can be linear or quadratic depending on the exact arrangement of the analyzer-polarizer angle. The apparent agreement unequivocally indicates that the weak quadratic term is responsible for the parabolic dependence on B-field and that the enhanced sensitivity of our technique is sufficiently high to detect it. We bring below this data (appearing in ‘Supplementary Information Note 2’).

Calculated and measured V_{PD} of Py for the B-modulation case for different analyzer-polarizer angles.

Calculated and measured V_{PD} of Py for the L -modulation case for different analyzer-polarizer angles.

The good agreement between the experiment and the calculation further indicates that the responses do not stem from some other artifact, but explicitly from the rotation of the polarization. A different effect that is quadratic in B -field should have prevented from obtaining the linear responses at the larger analyzer-polarizer angles. This is obviously not the case.

Magnetostriction may still take place but is expected to play a secondary role. For example, in the work by the Mak and Shan research group [Nat. Mat. **19**, 1295(2020)] it was shown that the mechanical and spin degrees of freedom were coupled in a suspended 2D magnetic membrane. In contrast, in our sample, the elasticity is significantly smaller since the film is much thicker and was deposited directly on a rigid substrate.

Finally, others have also measured the quadratic term. For example, in Li et. al [APL **124**, 054001 (2024), ‘Supplementary Information’ section S4], the standard perfectly cross-polarized configuration was implemented that is identical to our optical arrangement. In that setup, the extinction ratio was $> 10^5$ (without using an objective lens) and the quadratic dependence in Py was measured as well. In the figure below we bring the data from Li et. al (‘Supplementary Information’). Panel (a) presents θ_k and ϵ_k from which the measured signal, I_{sig} , can be determined and is presented in panel (b). I_{sig} readily reveals a quadratic dependence on B -field where a fit to a parabolic function results in $y = 0.1261x^2 + 0.0025$ with an R^2 determination value of 0.9774.

Measurement of quadratic term by Li et. al in a standard cross-polarized configuration ($\phi = 0^\circ$). (a) Kerr rotation (θ_k) and ellipticity (ϵ_k) taken from. (b) Reconstructed measured signal. $I_{sig} = I_0 \cdot |\theta_k + \epsilon_k|^2$. Solid line represents the quadratic fit resulting in a determination parameter of $R^2 = 0.9774$.

In the revised version of the manuscript, we have further elaborated on the ability to measure the quadratic term and its origin. To this end, the new ‘Supplementary Information Note 5’ was added. Additionally, we have pointed out the possible contribution of magnetostriction as reported by Jiang et. al in [Nat. Mat. **19**, 1295(2020)].

2) The cross-polarizer setup is widely used in the MOKE community, but almost always the linear response regime is used. Even in a 2D ferromagnet, which has a much lower MOKE response compared to 3D ferromagnets, a linear response, rather than a quadratic response, was used by intentionally rotating the analyzer angle from 90 degree (see Supplementary Section 2 of Nature Nanotechnology, 19, 1290 (2020)).

Indeed, in many cases the linear response is measured, especially when the signal is weak. However, this fact does not contradict the high sensitivity achieved when B -field is modulated that enables to operate in the quadratic-response regime. The quadratic regime is beneficial as it probes solely the pure MOKE response, unlike the linear regime, which results from intermixing with the primary polarization. Likewise, the noise figure is lower.

The ability to detect the quadratic term depends on the extinction ratio of the optical setup, the analyzer-polarizer arrangement, and of course on the sample itself which determines the strength of the magneto-optical effect. To show this, the signal should be compared to the detection shot noise. From Eq. (1) of the main text, for analyzer angles deviating by ϕ from maximal extinction, the detected photocurrent is given by:

$$I_{PD} \propto |E_s^i|^2 \cdot \left[\underbrace{\sin^2 \phi |r_{ss}|^2}_{\text{primary polarization, independent of } B} + 2 \cos \phi \sin \phi \cdot \underbrace{Re\{r_{ss}r_{ps}^*\}}_{\text{linear term in } B} + \cos^2 \phi \underbrace{|r_{ps}|^2}_{\text{quadratic term in } B} \right].$$

For FMs (@ 0.5 T): $|r_{ss}|^2 \sim 1$, $Re\{r_{ss}r_{ps}^*\} \sim 10^{-3} - 10^{-4}$, $|r_{ps}|^2 \sim 10^{-7} - 10^{-8}$,

while in NMs: $|r_{ss}|^2 \sim 1$, $Re\{r_{ss}r_{ps}^*\} \sim 10^{-4} - 10^{-5}$, $|r_{ps}|^2 \sim 10^{-8} - 10^{-10}$.

Since the primary polarization and the linear terms are orders of magnitude larger than the quadratic term, already for small ϕ angles, they will reach the detector and will immediately increase the shot noise.

Additionally, the finite extinction ratio, ER , of the polarizers allows the background light to reach the detector. To account for the finite ER , we followed a similar analysis as presented by Jin et. al in the Supplementary Section 2 of [Nat. Nano., **19**, 1290 (2020)] (derived in detail in a new 'Supplementary Information Note 5'). Accordingly, the optical intensity I_f reaching the detector can be described by $I_f = I_0|r_{ss}|^2 \left(\phi^2 + 2\phi \frac{|r_{ps}|}{|r_{ss}|} + \left(\frac{|r_{ps}|}{|r_{ss}|} \right)^2 + \frac{1}{ER} \right)$, where I_0 is the initial intensity of the laser.

The shot noise current, I_{NS} , can be calculated in a straight forward manner using $I_{NS} = \sqrt{2q \cdot BW_{det} \cdot \eta_{det} \cdot I_f}$ where BW_{det} is the bandwidth of the photodetector and η_{det} is its responsivity. The table below presents the calculated, I_{NS} , together with the detected photocurrent stemming from the quadratic and linear terms, I_{quad} and I_{linear} , respectively, for various ϕ angles and for different ER values, in typical FMs and NMs. The ER value of $3 \cdot 10^4$ and $\phi = 0.6^\circ$ appearing in the table was adopted from [Jin et. al, Nat. Nano., **19**, 1290 (2020)].

Our analysis indicates that in FMs the quadratic term can be detected for most configurations. However, in NMs, the quadratic term can be detected only for $\phi = 0^\circ$ (see 'Supplementary Information Note 5'), namely, at maximal extinction, and a high ER is required. Indeed, in the work by Li et. al mentioned above [APL **124**, 054001 (2024)], the cross polarized configuration was used while ER was $> 10^5$, and the quadratic response was observed in Py.

In our setup, Glan Thompson polarizers were used having $ER > 1:10^6$. In addition, B -modulation was applied which enhances the sensitivity such that eventually the quadratic term was measured even in NMs.

In 2D materials, such as those studied in the work by Jin et. al [Nat. Nano., **19**, 1290 (2020)] a ferromagnet was measured. Therefore, with the $ER > 3 \cdot 10^4$ of that work, presumably, it should be possible to detect the quadratic term. However, in atomically thin 2D magnets such as those described in Refs. [Nat. Materials **18**, 1303 (2019), Nat. Materials **19**, 838 (2020), Nano Lett. **21**, 5045 (2021), Nat. Nano. **17**, 143 (2022)] the magneto optic response is significantly weaker as compare to a 3D ferromagnet, resembling more closely the NM case. Namely, the quadratic term cannot be measured, and as the reviewer indicates, almost always the linear regime should be used.

ϕ [deg]	Material	Extinction ratio	I_{NS} [nA]	I_{quad} [nA]	I_{linear} [nA]
0°	FM	$3 \cdot 10^4$	4.61	97.4	-
		$1 \cdot 10^6$	0.8	97.4	-
	NM	$3 \cdot 10^4$	4.61	0.974	-
		$1 \cdot 10^6$	0.8	0.974	-
0.6°	FM	$3 \cdot 10^4$	10.04	97.4	2923.6
		$1 \cdot 10^6$	8.95	97.4	2923.6
	NM	$3 \cdot 10^4$	9.61	0.974	292.46
		$1 \cdot 10^6$	8.47	0.974	292.46

Comparison of shot noise and quadratic and linear terms of Eq. (1) of the manuscript. The table presents data for $\frac{|r_{sp}|}{|r_{ss}|} |_{0.5T}$ of $6.98 \cdot 10^{-4}$ and $6.98 \cdot 10^{-5}$ corresponding to typical values in FMs (Permalloy) and NMs having a response weaker by an order of magnitude. Red color coding indicates signals that are smaller than the shot noise and green indicates higher values. The ER value of $3 \cdot 10^4$ and $\phi = 0.6^\circ$ were used following Ref. [Nat. Nano. 19, 1290 (2020)]. More details are available in the new ‘Supplementary Information Note 5’.

To address the reviewer’s comment, in the revised version of the manuscript we have elaborated on the challenge of measuring 2D magnets and the new ‘Supplementary Information Note 5’ includes a detailed noise analysis. Further details of the setup were also added to the text.

3) To support authors’ drastic claim of the correlation between the noise level and spin-orbit coupling, I suggest a re-examination of their data in the linear response regime.

We turned to study the correlation after repeatedly observing that the heavy metals were much noisier as compared to the light metals: Pt and Ta were always noisy while the traces of Cu were significantly more stable. At first sight, the correlation with Gilbert damping may seem surprising. However, in light of the recent advances [PRR 6, 013012, (2024)], where the Gilbert damping was shown to determine the interaction strength with the magnetic component of light, it seems more reasonable.

Rotating the analyzer from maximal extinction will render the setup unsuitable for weak noise measurements for the following reasons:

1) The primary issue is that in the linear regime, the detection shot noise increases significantly. As shown above, when the analyzer is rotated away from maximal

extinction, the intense primary polarization and the intermixing components will reach the detector and increase the shot noise level.

2) In the linear regime, the detection shot noise induced by the significant linear term may affect the signal recorded by the lock-in amplifier. In this regime, the strong linear intermixing term will reach the detector. Its contribution to the shot noise is described

by: $I_{NS} \approx \sqrt{2 \cdot q \cdot BW_{det} \cdot I_0 \cdot \left(|r_{ss}|^2 \phi^2 + |r_{ps}| |r_{ss}| \phi + \frac{|r_{ss}|^2}{ER} \right)}$. Since r_{ps} is modulated by B , the intermixing shot noise term will also be modulated and may contaminate the signal recorded by the lock-in amplifier.

3) It is well known that the resolution of the measurement decreases as the magnitude of the measured signal increases. Since the noise seems to be independent of the magnetic field, it is understood to be additive to the magneto optic signal and independent of it. Therefore, in the linear regime, where magneto optic signal is much stronger, the measured voltages are orders of magnitude higher and the sensitivity to the small fluctuations is significantly reduced.

Actually, even if the noise depends on the magneto optic effect, it would be difficult to study the noise in the linear regime since the noise statistics in this case are influenced additionally by the cross-correlation between r_{ps} and r_{ss} whose nature is unknown. The cross-correlation stems from the intermixing term. In the quadratic regime, the pure r_{ps} term is measured and only the non-vanishing second moment contributes to the noise.

In the revised manuscript a new 'Supplementary Information Note 6' was added that includes a more detailed discussion of the noise measurements and addresses the differences between linear and quadratic regimes.

We thank the reviewer for their comments.

Response to comments of Reviewer #2

“I raised a concern about the validity of the quadratic relationship between the photodiode voltage and the B-field. The authors measured the photodiode voltage of the permalloy films at different deviation angles from the cross-polarization arrangement. In Fig. S2 (c), the measured photodiode voltage can be linear to the B-field at the deviation angle of 0.06 degree. At the deviation angle of 0 degree, the photodiode voltage is linear to the B-field at the low-field region and becomes nonlinear at the high-field region. However, this behavior cannot be described by a pure quadratic term. In my opinion, this result suggests that the pure quadratic response is almost impossible to achieve. Therefore, I cannot accept the relation between the square root of the photodiode voltage and the Kerr rotation. Considering this experimental concern and lack of theoretical analysis for the noise relation to spin-orbit coupling, I do not recommend the publication of this manuscript in this journal.”

We thank the reviewer for sharing their concern, however, we respectfully disagree with their opinion. The evidence presented in our earlier correspondence, including the demonstration that the response at angles between 0° and 0.06° closely aligns with the theoretical calculations, clearly support our understanding of the origin of the signal.

The revised manuscript now elaborates in greater detail on the possible origin of the connection between the noise and spin-orbit coupling as summarized below.

It is well known that the interaction between the optical radiation and spins in solids is mediated through SOC. In [PR **97**, 334 (1954)] Argyres considered the relativistic magnetic field that emerges from spin-orbit interaction upon the motion of the electron in the crystal field. In FMs, where the spin-up and spin-down populations are uneven, this strong effective magnetic field is responsible for the Kerr response. In NMs, where the spins are evenly distributed, the Kerr signal originates only from the Lorentz force induced by the external magnetic field which is spin-independent. However, each random spin still experiences a strong, spin-dependent relativistic field that affects its trajectory. This can induce the observed electromagnetic noise, which depends on the strength of the SOC.

Furthermore, there seems to be a link to the fluctuation-dissipation theorem. According to this theorem, magnetic fluctuations are inherently related to the mechanism responsible for the dissipation of spin angular momentum. In non-magnetic metals, the losses are mediated by spin-orbit coupling which transfers spin angular momentum to the lattice, as readily seen in spin pumping experiments (see Ref. [25], for example). Notably, the same spin-orbit interaction in the non-magnetic layer also governs the enhancement of Gilbert damping when coupled to a ferromagnet. Thus, possibly leading to the observed noise fluctuations.

Finally, we would like to point out that additional experiments were carried out on a new Al sample and at a different wavelength (638 *nm*). The results of these experiments were consistent with the conclusions of the work.

We appreciate the reviewer's effort in evaluating our work and would like to thank them for sharing their remarks.

Response to comments of Reviewer #3

“I have read the revised manuscript carefully, including all Supplement Notes, and the latest second-round rebuttal. I appreciate the authors’ great effort to improve the manuscript by providing detailed explanations of the reviewers’ concerns, which were added in the text of the manuscript and in additional Supplement Notes. However, I share the concerns of Referee #2, for the reasons given below, which make my opinion negative about the suitability of the manuscript for publication in Nature Communications in its current form.”

We would like to thank the reviewer for their careful evaluation of our work and for their thorough examination of the review process. Please find below our point-by-point response.

“I will divide my comments into Part A about the experimental results, and Part B about the interpretation.

A1. The authors determine the Kerr rotation values Φ_K as a function of the magnetic field B for a laser wavelength of $\lambda=440$ nm. They compare the measured Φ_K values for Cu, Au, Ta and Pt (Fig.4c) with those obtained for Py (Fig.2b). However, there is no basic comparison of the Φ_K values with the ones reported in the literature for the metals studied. For $\lambda=440$ nm, i.e. the photon energy ~ 2.8 eV, the measured Φ_K value in Ref. [20] is ~ 0.4 mdeg for Cu and ~ 0.4 mdeg for Au in an alternating magnetic field B of amplitude 1 T. Exactly corresponding values (0.6 mdeg and -0.6 mdeg, respectively) were measured in Ref. [10] in a constant field $B=1.5$ T. However, the Φ_K values measured in the manuscript for these metals are more than an order of magnitude higher (~ 3 mdeg at $B=0.45$ T for Cu and Au, Fig.4(c)) than the reported literature data!”

Following the reviewer’s remark, we conducted a detailed examination of the procedure applied to quantify Φ_K and concluded that the measurements were not calibrated properly. As a result, the $|\Phi_K|$ values were overestimated, as we explain below.

In addition to measuring the optical Hall effect (OHE) response $|r_{ps}|$, quantifying $|\Phi_K|$ requires determining the reflection coefficient $|r_{ss}|$ since $|\Phi_K| = \left| \frac{r_{ps}}{r_{ss}} \right|$. To this end, we positioned the photodetector directly after the sample. The signal was recorded using the same hardware used to measure $|r_{ps}|$ while applying an on-off modulation to the laser beam at $\omega_{mod} = 2\pi \cdot 700 \text{ Hz}$. In the process of re-examining our measurements, we realized that although the beam was attenuated, the temporal response of the photodetector was still highly distorted due to a combination of limited bandwidth and electrical saturation. As a result, the extracted $|r_{ss}|$ was smaller which ended up in larger $|\Phi_K|$ values.

It turns out that although the maximal bandwidth of the photodetector is $\sim 71 \text{ MHz}$ (Thorlabs DET36A2), it was inadvertently operated at a significantly reduced bandwidth of $\sim 397 \text{ Hz}$, below the modulation frequency of 700 Hz . The bandwidth is determined by considering the P-I-N junction capacitance, 40 pF in our case, and the load resistor. This reduction in bandwidth resulted from the large $10 \text{ M}\Omega$ input impedance of the lock-in amplifier (Stanford Research SR830) instead of using the recommended 50Ω load required to achieve maximal bandwidth. Panel (a) in the figure below shows the distorted photodetector signal, V_{PD}^0 , as measured with an oscilloscope, which ultimately led to the inaccurate evaluation of $|r_{ss}|$. Since the standard input impedance of an oscilloscope is only $1 \text{ M}\Omega$, to achieve the same $10 \text{ M}\Omega$ termination of the lock-in amplifier, a $9 \text{ M}\Omega$ resistor was connected in series to the input port of the oscilloscope. The signal appears saturated and clipped at the photodetector's battery supply voltage (12 V).

To remedy this issue, r_{ss} was measured again for all samples using a $100 \text{ K}\Omega$ load resistor which resulted in a bandwidth of $\sim 39.78 \text{ KHz}$ that eliminated the distortions as can be readily seen in Panel (b).

Temporal oscilloscope traces of the photodetector signal, V_{PD}^0 , in measurements of $|r_{ss}|$ with an on-off modulated laser beam at 700 Hz. (a) Acquired trace using a load of 10 M Ω , similar to the input impedance of the lock-in amplifier. (b) Acquired trace using a load of 100 K Ω .

In contrast to the measurements of r_{ss} , the OHE measurements of r_{ps} were not affected by the same saturation artefacts, but rather only by the reduced bandwidth. In principle, it is also desirable to measure r_{ps} using the extended bandwidth settings. However, since the load resistance also converts the photocurrent into voltage, increasing the bandwidth (by reducing the load resistance) causes the OHE signal to reduce proportionately. Given the vanishingly small OHE signal, this approach could not be applied in the measurement of $|r_{ps}|$. Therefore, to ensure that $|r_{ps}|$ was not affected by similar issues, the linear response of the photodiode was verified. In addition, the attenuation caused by limited bandwidth was quantified by measuring the frequency response.

The newly calibrated $|\Phi_K|$ values account for these details and were evaluated using the relation:

$$|\Phi_K| = \sqrt{\frac{V_{LI}^{ps}}{V_{LI}^{ss}}} \cdot \sqrt{\frac{R_L^{ss}}{R_L^{ps}} \cdot \frac{L_{OD}^{ss}}{L_{BW}^{ps}} \frac{\mathcal{F}_{rect}}{\mathcal{F}_{sine}}}$$

where V_{LI}^{ps} and V_{LI}^{ss} are the recorded lock-in amplifier voltages (upper indices indicate the r_{ps} - and r_{ss} -related quantities) corresponding to $V_{PD,B_{mod}}$ and V_{PD}^0 in the manuscript, respectively. R_L^{ss} and R_L^{ps} are the load resistance used in the experiments, L_{OD}^{ss} is the

optical attenuation applied in the measurement of $|r_{ss}|$, L_{BW}^{ps} is the attenuation resulting from the limited bandwidth in the measurement of $|r_{ps}|$, and \mathcal{F}_{rect} and \mathcal{F}_{sine} are the Fourier components at ω_{mod} of the normalized square and the all-positive sinusoidal, $\frac{1}{4} \cdot (1 + \sin(\omega_{mod}t))^2$, modulation waves applied in the measurements of $|r_{ss}|$ and $|r_{ps}|$, respectively.

The table below presents a comparison between the newly obtained and previously reported $|\Phi_K|$ values in the literature at 440 nm. The table presents the extrapolated values to the maximal field applied in our experiments (0.45 T). The table also includes data for Al which was not part of the original dataset and is discussed in the following point. The values are in relatively good agreement with those reported in Refs. [10, 19, 20, 37] of the revised manuscript. The differences may be attributed to the different film growth conditions and film thicknesses. For example, Uba et. al. in Refs. [10, 37] measured 200 nm thick films versus 50 nm in our work.

Sample	Present work	Ref. [10], Uba et al. (2017) Ref. [37], Uba et al. (2000)		Ref. [20] Schnatterly et al. (1969)		Ref. [19] Stern et al. (1964)	
	Measured @ 0.45 T [mdeg]	Measured @ 1.5 T [mdeg]	Extrapolated to 0.45 T [mdeg]	Measured @ 1 T [mdeg]	Extrapolated to 0.45 T [mdeg]	Measured in units of [mdeg/T]	Extrapolated to 0.45 T [mdeg]
Au	$ \Phi_K = 0.39$	$ \Phi_K = 1.18$	$ \Phi_K = 0.33$	$ \Phi_K = 0.61$	$ \Phi_K = 0.27$	-	-
Cu	$ \Phi_K = 0.33$	$ \Phi_K = 0.84$	$ \Phi_K = 0.28$	$ \Phi_K = 0.66$	$ \Phi_K = 0.29$	-	-
Pt	$ \Phi_K = 0.47$	$ \Phi_K = 0.1$	$ \Phi_K = 0.3$	-	-	-	-
Al	$ \Phi_K = 0.22$	-	-	-	-	$\theta_k = 0.4$	$\theta_k = 0.18$
Ta	$ \Phi_K = 0.37$	-	-	-	-	-	-

Comparison of measured and previously reported $|\Phi_K|$ values at 440 nm.

As a result of the new calibration, Fig. 2(b) and Fig. 4(c) were updated. The procedure applied to extract $|\Phi_K|$ is now described in detail in Supplementary Note 1 while additional information regarding the detector and lock-amplifier was added to the 'Methods' section. Finally, the table describing the quantitative comparison was added to Supplementary Note 1 as well and the main text was updated accordingly.

"A2. The fundamental discrepancy between the measured large $|\Phi_K|$ values and the literature data strengthens concerns, already expressed by Referee #2, about the origin of the measured signal, that a mechanism other than Kerr rotation may be

responsible for the measured Φ_K . To resolve this discrepancy, my recommendation is that the authors should measure and compare the Kerr rotation on the Al film, for which the Φ_K at 0.45 T, estimated from the data measured and reported in the literature for Al metal (Ref. [19]), is smaller (0.18 mdeg for $B=0.45$ T), and the plasma frequency is by ~50 % higher than for Cu and Au.”

Following the reviewer’s recommendation, an additional Al sample was measured which indeed had a weaker response. To this end, we grew a 50 nm thick Al film which was capped with 2 nm of TaN to prevent natural oxidation and additional XRD, AFM, and DC Hall characterization were carried out to verify the film quality.

This new OHE measurement, together with the previously measured films, is presented below in the updated Fig. 4. Panel (a) presents the measured $V_{PD,B_{mod}}$ illustrating the expected quadratic dependence on B_{AC} and that the response of Al is weakest. Panel (b) presents $\sqrt{\tilde{V}_{PD}}$ together with the results of the Lorentz-Drude model. It is seen that also $\sqrt{\tilde{V}_{PD}}$ of Al is linear in B . The Lorentz-Drude model predicts distinct responses for Pt and Ta, setting them apart from those of Au, Cu, and Al due to the relatively higher plasma frequencies of Au, Cu, and Al. However, in the measurements, the responses are more similar. This discrepancy may be attributed to 3d and 5d interband transitions, which cause deviations from the model, an effect highlighted by the reviewer and addressed in the following comment. Panel (c) presents the extracted $|\Phi_K|$ data. The measured $|\Phi_K|$ value of 0.22 mdeg at 450 mT is comparable to the value reported by Stern et al. in Ref. [19] (0.18 mdeg).

Ferris MOKE measurements of Cu, Au, Al, Ta, and Pt. (a) Measured $V_{PD,B_{mod}}$. Dashed lines represent the parabolic fit. (b) Measured $\sqrt{\tilde{V}_{PD,B_{mod}}}$ vs. B_{AC} (solid lines). Dashed lines represent the linear fit of $\sqrt{\tilde{V}_{PD,B_{mod}}}$. Close triangles represent the theoretically calculated $\sqrt{\tilde{V}_{PD}}$. (c) $|\Phi_K|$ of Pt, Ta, Al, Au, and Cu. (d) Extracted $\sqrt{\tilde{V}_{Noise}}$ traces as a function of B_{AC} . (e) $\sqrt{\tilde{V}_{RMSE,norm}}$ as a function of

$\alpha_{B_{mod}}^{1/2}$. (f) $\sqrt{\tilde{V}}_{RMS,norm}$ as a function of α_{sp}^{YIG} . Values were taken from Ref. [25]. Inset: Noise-damping correlation in Py/NM bilayers. In (e) & (f) black dashed line represents a linear fit. In all panels, traces of Cu, Au, Al, Ta, and Pt are indicated by red, yellow, grey, green, and blue colors, respectively.

It is particularly interesting to examine whether the correlation between the Gilbert damping enhancement, $\alpha_{sp,Al}^{Py}$, and noise level also persists in the Al sample. Therefore, spin pumping measurements were also carried out using the OSTFMR setup. To this end an additional sample consisting of 50 Al | 10 Py | 2 TaN (thicknesses in nm) was fabricated and devices were patterned. These measurements yielded $\alpha_{sp,Al}^{Py}$ of 2.7×10^{-3} . The slightly higher α_{sp} of Al as compared to Cu is consistent with the previously measured data [Rothschild, PRB, **106**, 144415]. Panel (d) presents the noise signal $\sqrt{\tilde{V}}_{Noise}$ obtained by subtracting the linear fit from $\sqrt{V_{PD}}$. Panel (e) presents the RMS noise, $\sqrt{\tilde{V}}_{RMS,norm}$, as a function of the magnitude of the optical Hall response, $\alpha_{B_{mod}}^{1/2}$. No observable correlation is seen, consistent with our earlier conclusion. In contrast, a good correlation between the noise and $\alpha_{sp,Al}^{Py}$ is readily seen, as presented in the inset of Panel (f). The additional data point of Al is consistent with the conclusion that the noise scales with α_{sp} .

The revised manuscript now includes the results for Al which are presented in the updated Fig. 4, and the main text was modified accordingly. Tables 1 and 2 were updated as well and now summarize the relevant data for the Al film. Additionally, Supplementary Note 5 now includes the XRD, AFM, and DC Hall characterization of the Al film. Finally, the OSTMR damping measurements of Al were added to Supplementary note 8.

“B1. In line 175 of the revised manuscript, the following sentence is written “Core-level transitions are neglected as they occur in the X-ray range”. This statement is true, but the authors forgot that 3d and 5d interband transitions cannot be neglected. As it is well known, at such a high photon energy as 2.8 eV, the interband transitions are dominant and masking the free-electron-contribution intraband transitions (see, e.g., Refs. [20] and [10]).”

The reviewer raises an important point that indeed should have been noted in the earlier versions. As evident from Fig. 4(b), the Lorentz-Drude model is incomplete as it accounts only for the intraband transitions. In Ref. [10] by Uba et. al. (2017), it was shown that when the 3d and 5d interband transitions are considered, e.g. in Cu and Au, the calculated magneto-optic spectra closely reproduce the experimental measurements. The same point was discussed, though with brevity, also in the earlier work by Schnatterly (1969) in Ref. [20] for Au, and in [Uba et al., PRB **62**, 16510 (2000)] for Pt. For example, in Cu, it was shown that the interband transitions take place in the vicinity of $\sim 2.1 - 2.4 \text{ eV}$ corresponding to $\sim 517 - 590 \text{ nm}$. This is likely to explain the observed deviation between the theory and measurements of Fig. 4(b).

The revised version of the manuscript now discusses the contribution of the interband transitions.

“B2. Why did the authors choose a blue laser as the light source? Using a red or infrared laser would allow measurement below the energy edge of interband transitions, i.e., only the free-electron contribution in the studied metals”

The blue laser was primarily chosen for technical reasons.

Observing features related to the energy band edge would indeed be interesting. Therefore, following the reviewer’s remark, we repeated the measurements with a red 638 nm laser (Cobolt 06-MLD, 10 mW, beam size: $700 \times 700 \mu\text{m}^2$). The results are presented in the figure below. As compared to 440 nm, the Kerr rotation angles at 638 nm are smaller. Panel (a) shows the measured $V_{PD,B_{mod}}$ exhibiting the clear quadratic dependence. At this wavelength a different ordering of the responses is observed as compared to 440 nm. The response is strongest for Au, followed by Cu, Ta, Pt, and then Al, suggesting distinct spectral features in the range of 1.9–2.8 eV. Panel (b) displays the Kerr rotation angles, which vary linearly with B_{AC} and are overall consistent with previously reported values, as summarized in the table below. The smaller $|\Phi_K|$ values may indicate a smaller contribution of the interband transitions. Panels (c) and (d) present the noise analysis. In these measurements, the lack of correlation between $\sqrt{\tilde{V}_{RMSE,norm}}$ and the optical Hall response is even more pronounced, for example, Au and Cu exhibit the largest responses but the lowest noise

levels and Pt displays a weak response that is accompanied by significant noise. Once again, as shown in Panel (d), the noise scales with α_{sp} resulting in an R-square determination parameter of 0.915.

(a) $V_{PD,B_{mod}}$ measured at 638 nm. (b) Extracted $|\Phi_K|$ values. (c) Extracted $\sqrt{V_{Noise}}$ traces as a function of B_{AC} . (d) $\sqrt{V_{RMS,norm}}$ as a function of α_{sp}^{Py} .

Sample	Present work	Ref. [10], Uba et al. (2017) Ref. [37], Uba et al. (2000)		Ref. [20] Schnatterly et al. (1969)		Ref. [19], Stern et al. (1964) Sup. Ref. [18], McGroddy et. al (1965)	
	Measured @ 0.45 T [mdeg]	Measured @ 1.5 T [mdeg]	Extrapolated to 0.45 T [mdeg]	Measured @ 1 T [mdeg]	Extrapolated to 0.45 T [mdeg]	Measured in units of [mdeg/T]	Extrapolated to 0.45 T [mdeg]
Au	$ \Phi_K = 0.36$	$ \Phi_K = 1.26$	$ \Phi_K = 0.42$	$ \Phi_K = 0.81$	$ \Phi_K = 0.36$	$\theta_k = 0.8$	$\theta_k = 0.36$
Cu	$ \Phi_K = 0.32$	$ \Phi_K = 0.61$	$ \Phi_K = 0.2$	$ \Phi_K = 0.28$	$ \Phi_K = 0.13$	-	-
Pt	$ \Phi_K = 0.18$	$ \Phi_K = 0.28$	$ \Phi_K = 0.085$	-	-	-	-
Al	$ \Phi_K = 0.17$	-	-	-	-	$\theta_k = 0.31$	$\theta_k = 0.15$
Ta	$ \Phi_K = 0.26$	-	-	-	-	-	-

Comparison of measured and previously reported $|\Phi_K|$ values at 638 nm.

The results using the red laser are now presented in the new Supplementary Note 9 and are described briefly in the main text.

“In conclusion, the last statement in the abstract of the manuscript “These results suggest that the electromagnetic noise arises from optical interactions with the spins and highlight a possible avenue for measuring the spin-orbit interaction using optical techniques” requires, in my opinion, more justification, both experimental and theoretical. Additional test measurements on Al reference sample, for which Kerr rotation is small, plasma frequency is high, and spin-orbit interaction is from one to two orders of magnitude smaller than for the metals studied, are highly recommended.”

As mentioned above, the revised manuscript includes additional experimental justifications for the noise correlation which include the measurements using the red laser and the additional Al sample. Additionally, a detailed discussion regarding the possible noise mechanisms was added which is summarized below.

It is well known that the interaction between the optical radiation and spins in solids is mediated through SOC. In [PR **97**, 334 (1954)] Argyres considered the relativistic magnetic field that emerges from spin-orbit interaction upon the motion of the electron in the crystal field. In FMs, where the spin-up and spin-down populations are uneven, this strong effective magnetic field is responsible for the Kerr response. In NMs, where the spins are evenly distributed, the Kerr signal originates only from the Lorentz force induced by the external magnetic field which is spin-independent. However, each random spin still experiences the strong, spin-dependent relativistic field that affects its trajectory. This can induce the observed electromagnetic noise fluctuations, which depend on the strength of the SOC.

The correlation between the noise and α_{sp} seems also to be consistent with the fluctuation-dissipation theorem. According to this theorem, magnetic fluctuations are inherently related to the losses of spin angular momentum to the lattice which are manifested by the Gilbert damping, α . Accordingly, the magnetic fluctuations are given by: $S(\omega) = \frac{2k_B T}{\omega} \chi''(\omega)$, where $\chi''(\omega)$ is the dissipative part of the response function, k_B is Boltzmann’s constant, T is the temperature, and ω is the frequency. In magnetic systems, χ'' is proportional to α . In non-magnetic metals, the losses are mediated by spin-orbit coupling which transfers spin angular momentum to the lattice, as readily seen in spin pumping experiments (see Ref. [25], for example). Therefore, the analogous parameter to α in non-magnetic metals is the spin-orbit coupling strength,

ξ_{NM} . Notably, the same spin-orbit interaction in the non-magnetic layer also governs the enhancement of the Gilbert damping, α_{sp} , when coupled to a ferromagnet. This may explain the observed noise behavior (further details are provided in the revised manuscript).

We thank the reviewer for their valuable feedback. We believe that the amendments made following their remarks have improved the quality of the manuscript. We hope the reviewer finds the revised manuscript satisfactory and that it adequately addresses their concerns.

Response to comments of Reviewer #4

As the manuscript was already reviewed by other referees, one of the issues given by the other referee is validity of quadratic dependence of MOKE signals with respective to magnetic field. I have read authors rebuttal and found that such quadratic dependence is natural in the perfect cross-polarizer configuration and also reported in the previous study [Li et al. Appl. Phys. Lett. 124, 054001 (2024)] as authors mentioned. The analyzer angle was slightly rotated to detect linear magnetic response in the most of previous works.

We thank the reviewer for carefully examining our responses and the previously reported data.

However, I also found a thing different with the point mentioned by the other referee that the measurement technique reported in the present manuscript is different with the previous study, which might cause the confusion. Authors used lock-in amplifier to detect modulated MOKE signals that is proportional to square of Kerr rotation or magnetic field. That is,

$$r_{ps} \propto \sin(\omega_{mod}t), r_{ps}^2 \propto \sin^2(\omega_{mod}t) = \frac{1}{2}(1 + \cos(2\omega_{mod}t))$$

So, the signal measured at the photo-diode is proportional to the signal with the frequency twice larger than the modulated signal and the dc signal. I feel that this fact should be clarified in the revised manuscript.

The most straightforward description of the modulation is indeed a sine wave. However, since the magnetic field goes to zero between magnets rather than being negative, an “all-positive” sine wave is the more appropriate description. Accordingly,

$$r_{ps} \propto 1 + \sin(\omega_{mod}t) \text{ and } r_{ps}^2 \propto (1 + \sin(\omega_{mod}t))^2 = \frac{3}{2} + 2 \sin(\omega_{mod}t) - \frac{1}{2} \cos(2\omega_{mod}t)$$

which is still modulated at ω_{mod} . This is illustrated visually in the figure below by plotting the functions $f = 1 + \sin(\omega_{mod}t)$ and $f = (1 + \sin(\omega_{mod}t))^2$ which shows that the primary modulation of r_{ps}^2 remains at the fundamental frequency.

Visual illustration of the modulations of r_{ps} and r_{ps}^2 . Red solid line: $f(t) = 1 + \sin(\omega_{mod}t)$. Blue solid line: $f(t) = (1 + \sin(\omega_{mod}t))^2$.

In accord with the reviewer's remark, r_{ps}^2 should also contain a component at $2\omega_{mod}$, with a magnitude one-fourth that of the signal at ω_{mod} . To experimentally demonstrate this signal, additional measurements were performed on the Py film where the reference frequency was set to ω_{mod} and $2\omega_{mod}$. These measurements are presented below. In each measurement, the 'in-phase' (X) and 'quadrature' (Y) components of the lock-in amplifier were recorded. The results show that the signal at $2\omega_{mod}$ exhibits a quadratic dependence on B_{AC} , with an amplitude approximately one-quarter of the signal at ω_{mod} . Furthermore, while the signal at ω_{mod} appears in the 'in-phase' (X) component, the signal at $2\omega_{mod}$ is observed in the 'quadrature' (Y) component of the lock-in amplifier, as expected of a cosine-modulated signal.

Measured response of Py at the fundamental and second harmonics. Green line: measured signal at ω_{mod} . Yellow line: measured signal at $2\omega_{mod}$. Black dashed lines represent a fit to a quadratic function.

This point is now better explained in the revised manuscript and to improve clarity, the modulated magnetic field, B_z , is now expressed in terms of the “all-positive” sine wave $B_z = b_0(l) + \frac{1}{2}B_{AC}(l) \cdot (1 + \sin(\omega_{mod}t))$ (instead of $B_z(t, l) = B_0(l) + \frac{1}{2} \cdot B_{AC}(l) \cdot \sin(2\pi f_{mod} \cdot t)$ appearing originally) where $b_0(l)$ expresses a small deviation from the ideal “all-positive” sine wave. Additionally, a new Supplementary Note 2 was added which demonstrates the measurements at ω_{mod} and $2\omega_{mod}$.

“Another concern about the work I think is that the physical mechanisms of the correlation between noise level and spin-orbit coupling in nonmagnetic metals are not discussed in detail in the present manuscript.

For example, Gilbert damping parameter and thermal fluctuation of magnetization are related due to the fluctuation-dissipation theorem. I think that the noise in nonmagnet may be caused by the following mechanism; the spin induced by magnetic field (Pauli susceptibility) in nonmagnet has particular life-time, namely spin relaxation time which

should be related to spin-orbit coupling of the material according to Elliot-Yafet mechanism. Then, this dissipation is equal to thermal fluctuation of spin.

Authors should describe the mechanism of noise observed in nonmagnetic metals in the revised manuscript. I think that discussions in the present manuscript rely on the Gilbert damping which is the parameter for ferromagnet. However, authors measure the noise without ferromagnet, which contradicts the discussion presented.

In addition to the revision of discussions, temperature dependent measurement might provide more insights into the mechanism if possible.”

The Gilbert damping parameter, α , refers to magnetic systems rather than nonmagnetic (NM) systems. It reflects the ability to transfer spin angular momentum to the lattice. The corresponding parameter in NMs is the spin orbit coupling (SOC) parameter, ξ_{NM} . In thin FMs films, it was shown, that α is proportional to ξ_{FM}^2 , where ξ_{FM} is the SOC parameter of the FM [Kamberský, Czech. J. Phys. B, **26**, 1366 (1976); He et al., PRL **110**, 077203 (2013)]. α_{sp} which describes the enhancement of the Gilbert damping of a FM due to spin pumping when an NM is introduced to the magnetic system is fundamentally related to ξ_{NM} . It originates from the dissipation of spin angular momentum in the NM but is expressed in terms of losses of a magnetic system. As the reviewer noted, the mechanism responsible for the dissipation of spin angular momentum in the NMs that is relevant in our experiments is the spin relaxation by the Elliott–Yafet mechanism. Within this framework, spin-flip accompany electron scattering events such that the spin relaxation time, τ_s , is related to τ according to $\tau_s^{-1} \propto \xi_{NM}^2 \cdot \tau^{-1}$ [Boross et al., Sci. Rep. 3, 3233 (2013)]. This is the same dependence on ξ_{NM} that was found to describe α_{sp} in FM/NM bilayers [Barati et al., PRB **90**, 0144202014 (2014)].

It seems that the fluctuation–dissipation theorem leads to a similar conclusion. Accordingly, the power spectral density of the spontaneous fluctuations can be expressed by $S(\omega) = \frac{2k_B T}{\omega} \chi''(\omega)$, where $\chi''(\omega)$ is the dissipative part of the response function, k_B is Boltzmann’s constant, T is the temperature, and ω is the frequency. Therefore, in magnetic systems, $S(\omega)$ is proportional to α . Analogously, it seems plausible that in NMs the spin fluctuations would be proportional to α_{sp} .

Generally, it is well known that the interaction between the optical radiation and spins in solids is mediated through SOC. In [PR **97**, 334 (1954)] Argyres considered the relativistic magnetic field that emerges from spin-orbit interaction upon the motion of the electron in the crystal field. In FMs, where the spin-up and spin-down populations are uneven, this strong effective magnetic field is responsible for the large Kerr response. In NMs, where the spins are evenly distributed, the Kerr signal originates only from the Lorentz force induced by the external magnetic field which is spin-independent. However, each random spin still experiences a strong, spin-dependent relativistic field that affects its trajectory. This can induce the observed electromagnetic noise fluctuations, which depend on the strength of the SOC.

Finally, we would like to note that it is unclear whether the spin polarization induced by the magnetic field contributes to the noise mechanism as the noise does not scale with B_z .

The temperature-dependent measurement suggested by the reviewer could indeed lead to a better understanding of the noise mechanism. However, such measurements are not currently supported by our experimental setup.

To prevent possible confusion, we have revised the terminology used in the relevant sections. A discussion of the relation between α , ξ , and α_{sp} has been added to the manuscript for clarity. In addition, we have added a detailed discussion addressing the possible physical origins of the noise mechanism. Finally, we have pointed out the temperature-dependent measurements as a particularly important avenue for a future study.

We would like to thank the reviewer for their valuable remarks and for the evaluation of our work.

Response to comments of Reviewer #3

“The authors have adequately addressed my concerns and I find the revised manuscript satisfactory. In my opinion, the manuscript in the present form is suitable for publication in Nature Communications.”

We would like to thank the reviewer for their careful evaluation of our work and for their efforts in assessing our work.

Response to comments of Reviewer #4

“Authors modified the manuscript and addressed the concerns according to my previous review. I think that they modified the manuscript properly and the manuscript is now acceptable to be published.”

We would like to thank the reviewer for their valuable remarks and for the evaluation of our work.